# Water induced ultrathin Mo$_2$C nanosheets with high-density grain boundaries for enhanced hydrogen evolution

Yang Yang[1,2,12], Yumin Qian[3,12], Zhaoping Luo[4], Haijing Li[5], Lanlan Chen[6], Xumeng Cao[4], Shiqiang Wei[6], Bo Zhou[7], Zhenhua Zhang[8], Shuai Chen[9], Wenjun Yan[9], Juncai Dong[5], Li Song[6], Wenhua Zhang[6], Renfei Feng[10], Jigang Zhou[10], Kui Du[4], Xiuyan Li[4], Xian-Ming Zhang[1,2] ✉ & Xiujun Fan[1,11] ✉

Grain boundary controlling is an effective approach for manipulating the electronic structure of electrocatalysts to improve their hydrogen evolution reaction performance. However, probing the direct effect of grain boundaries as highly active catalytic hot spots is very challenging. Herein, we demonstrate a general water-assisted carbothermal reaction strategy for the construction of ultrathin Mo$_2$C nanosheets with high-density grain boundaries supported on N-doped graphene. The polycrystalline Mo$_2$C nanosheets are connected with N-doped graphene through Mo−C bonds, which affords an ultra-high density of active sites, giving excellent hydrogen evolution activity and superior electrocatalytic stability. Theoretical calculations reveal that the $d_z^2$ orbital energy level of Mo atoms is controlled by the MoC$_3$ pyramid configuration, which plays a vital role in governing the hydrogen evolution activity. The $d_z^2$ orbital energy level of metal atoms exhibits an intrinsic relationship with the catalyst activity and is regarded as a descriptor for predicting the hydrogen evolution activity.

Owing to the high gravimetric specific energy density and environmentally friendly characteristics, hydrogen energy has emerged as one of the most promising alternatives to fossil fuels[1,2]. Electrochemical hydrogen evolution reaction (HER) provides an attractive way for efficient H$_2$ production from water electrolysis, in which catalysts are critical for developing renewable energy conversion technologies[3]. To date, platinum (Pt)-based materials are considered to be the most active HER catalysts, but the scarcity and high cost of Pt rigorously hamper their widespread applications[4]. Therefore, developing earth-abundant alternatives to Pt-based catalysts for achieving efficient H$_2$ generation is highly desirable. Molybdenum carbide (Mo$_2$C), an excellent early transition-metal carbide, has intensely

[1]Institute of Crystalline Materials, Shanxi University, Taiyuan, Shanxi 030006, China. [2]Key Laboratory of Interface Science and Engineering in Advanced Materials, College of Chemistry, College of Materials Science and Engineering, Taiyuan University of Technology, Taiyuan, Shanxi 030024, China. [3]Beijing Key Lab of Nanophotonics and Ultrafine Optoelectronic Systems, School of Physics, Beijing Institute of Technology, Haidian, Beijing 100081, China. [4]Shenyang National Laboratory for Materials Science, Institute of Metal Research, Chinese Academy of Sciences, Shenyang 110016, China. [5]Beijing Synchrotron Radiation Facility, Institute of High Energy Physics, Chinese Academy of Sciences, Beijing 100049, China. [6]National Synchrotron Radiation Laboratory, University of Science and Technology of China, Hefei, Anhui 230026, China. [7]Institute of Microstructure and Properties of Advanced Materials, Beijing University of Technology, Chaoyang District, Beijing 100124, China. [8]Innovative Center for Advanced Materials, Hangzhou Dianzi University, Hangzhou, Zhejiang 310018, China. [9]Institute of Coal Chemistry, Chinese Academy of Sciences, Taiyuan 030001, China. [10]Canadian Light Source, Saskatoon, SK S7N2V3, Canada. [11]Engineering Research Center of Energy Storage Materials and Devices, Ministry of Education, School of Chemistry, Xi'an Jiaotong University, Xi'an 710049, China. [12]These authors contributed equally: Yang Yang, Yumin Qian. ✉e-mail: zhangxm@dns.sxnu.edu.cn; fxiujun@gmail.com

awakened ever-growing interest as a promising HER electrocatalyst because its electronic structure is virtually analogous to that of Pt metal[5,6]. Nevertheless, $Mo_2C$ surface shows excessively strong Mo–H binding energy, which hinders the desorption of adsorbed H to generate $H_2$, severely deteriorating the electrochemical HER activity[7].

Grain boundaries (GBs), as a type of planar defect, are effective to directly tune the surface atomic and electronic structure, significantly altering the intrinsic reactivity of nanocrystalline materials[8,9]. Specifically, GBs create lattice distortion regions in $Mo_2C$ polycrystals by stabilizing dislocations, which could provide an optimized electronic structure for tailoring the binding energy of Mo–H, consequently accelerating the $H_2$ production. However, conventional bulk materials are typically limited to the surface GB density, which is not available for practical catalytic reactions due to the relatively low accessibility of active sites. Compared with bulk counterparts, ultrathin two-dimensional (2D) nanostructures possess sufficient interplanar active sites and shorter reactant/product diffusion length, and have been intensively reported as promising candidates for HER[10]. Currently, most of the $Mo_2C$ nanostructures are synthesized from either high-temperature carburization of costly/toxic molybdenum precursors (e.g., Mo foil, $MoF_6$, and $Mo(CO)_6$) with carbonaceous gases (e.g., $CH_4$, $C_2H_6$, and CO)[7,11] or the selectively etching layered ternary Mo-containing phases[12,13]. Nevertheless, in the former method, the resultant carbide surface generally suffers from severe char contamination from the pyrolysis of carbonaceous gases, inhibiting the exposure of active sites; in the latter, the delamination yield and defect controllability of $Mo_2C$ sheets derived in aqueous fluoride-containing acidic solutions are not satisfactory. In this context, nearly all the above-mentioned methods can only introduce small changes in the structural and chemical configuration of $Mo_2C$ catalysts, resulting in poor tunability for altering the catalytic performance. Therefore, a facile approach for developing high-density and fully exposed GBs on ultrathin 2D $Mo_2C$ nanostructure is highly desired.

Herein, we propose an effective strategy to achieve ultrathin $Mo_2C$ nanosheets (NSs) with rich GBs supported on N-doped graphene (H-$Mo_2C$/NG) via hydrothermal and water-assisted carbothermal reactions. During the carbonization process, water induces the structural evolution of $Mo_2C$ nanocrystals from nanoparticles (NPs) to NSs and controls the GB density of $Mo_2C$ NSs as well. The high-density GBs in $Mo_2C$ NSs provide an ultra-high fraction of active sites, significantly improving the inherent HER activity of H-$Mo_2C$/NG. Theoretical calculations show that the GBs in $Mo_2C$ NSs modulate the configuration of $MoC_3$ pyramid, which thereby regulates the Mo $d_z^2$ orbital energy level, manipulating the Mo–H bond strength of H-$Mo_2C$/NG catalyst and influencing the HER activity. This work opens up an avenue for the development of high-efficiency catalysts through GB engineering.

## Results

### Controllable construction of ultrathin $Mo_2C$ NSs with high-density GBs

$Mo_2C$ NSs supported on N-doped graphene (NG) were constructed in situ by facile hydrothermal and water-assisted carbothermal reactions (Fig. 1a). Typically, graphene oxide (GO) and $(NH_4)_6Mo_7O_{24}\cdot4H_2O$ were dissolved into deionized water and then the homogeneous solution was subjected to a hydrothermal reaction to obtain a homogeneous mixture composed of amorphous $MoO_3$ supported on reduced GO sheets ($MoO_3$/RGO) (Supplementary Fig. 1)[7]. Next, the mixture was freeze-dried and then treated at 800 °C under $NH_3$/Ar gas. Finally, only $Mo_2C$ NPs decorated on NG were obtained when the $MoO_3$/RGO intermediate was water-free. However, when $MoO_3$/RGO intermediate contained sufficient water concentration (18.56 wt%), $Mo_2C$ NSs with high-density GBs were constructed in situ and anchored on NG. Further details of the experiments were provided in Methods.

To illustrate the growth behavior of $Mo_2C$ nanocrystals with the assistance of water, density functional theory (DFT) calculations were performed. As shown in Fig. 1b, two theoretical model systems consisting of initial $MoO_3$ without $H_2O$ ($IS_{NP}$) and with $H_2O$ ($IS_{NS}$) are constructed, and the structural evolution mechanisms of $Mo_2C$ products are discussed from a reaction enthalpy ($\Delta H$) viewpoint. During the carbonization process, $IS_{NP}$ and $IS_{NS}$ first absorb heat and dissociate into transition states of $TS_{NP}$ and $TS_{NS}$, respectively, and they then react with C atoms in RGO to form $Mo_2C$. Generally, RGO has two types of C atoms, namely, the highly reactive carbon at the edge and the relatively inertness carbon of the six-membered C-ring far away from the edge[14]. Because H-bonded graphene has a structure analogous to benzene (Ph), the chemical properties of the compounds can be thought to be similar[15]. For simplicity, the C atoms extracted from the edge of the graphene lattice, as well as the free Ph radicals, react with $TS_{NP}$ and $TS_{NS}$ to produce $TS_{NP}$-C, $TS_{NP}$-Ph, $TS_{NS}$-C, and $TS_{NS}$-Ph, respectively. Of course, these model structures are much simpler than those in the actual experiment, yet, they are sufficient to study the interaction between two substances. Theoretical calculations show that $\Delta H$ values for $TS_{NP}$ and $TS_{NS}$ reacting with C atoms are both negative, which indicates that the C atoms from RGO can carbonize $MoO_3$ to form $Mo_2C$. In anhydrous atmosphere, C atoms are chemically intercalated into common $MoO_3$ motifs, reducing $MoO_3$ into $Mo_2C$ NPs decorated on defective NG. In addition, the reaction $\Delta H$ of $TS_{NP}$ to produce $TS_{NP}$-Ph is 0.45 eV, suggesting that $TS_{NP}$ fails chemically bind with the six-membered C-ring. But the reaction $\Delta H$ for $TS_{NS}$ to produce $TS_{NS}$-Ph is −1.45 eV, which indicates that $TS_{NS}$ tends to combine with six-membered C-ring on RGO basal plane, thus inducing the lateral arrangement of $Mo_2C$ lattice along 2D direction[16]. In general, common $MoO_3$ without $H_2O$ can only react with carbon on RGO edge, producing $Mo_2C$ NPs. While $MoO_3$ with $H_2O$ not only reacts with the edge C atoms but also readily combines with six-membered C-ring along the RGO basal plane; in this situation, epitaxial growth is carried out to enlarge the $Mo_2C$ domain, achieving a coalesced NSs with several micrometers in lateral size. Especially, in the presence of $H_2O$, due to the guest–host interaction between $H_2O$ and $MoO_3$[17], a unique Mo–O configuration is obtained and anchored on RGO, which serves as the nucleation site for $Mo_2C$ growth. As the activated carbon species encounter the Mo–O configuration, the dense seeds begin to individually nucleate at multiple regions. Consequently, these simultaneously grown nanocrystals merge together seamlessly and eventually form multi-faceted $Mo_2C$ NSs, which thereby hinders the growth of $Mo_2C$ grains in the radial orientation, leaving behind numerous GBs. In this situation, the RGO intermediate not only provides C source for $Mo_2C$ growth, but also acts as a robust supports for $Mo_2C$ nanodomains to nucleate and finally splice into NSs; in turn, the formed $Mo_2C$ NSs are covalently connected with underneath NG sheets through Mo–C bonds, which ensures the high inherent stability of $Mo_2C$ hybrids.

Morphologies and structures of the $Mo_2C$ hybrids were characterized with atomic force microscopy (AFM), transmission electron microscopy (TEM), and high-angle annular dark-field scanning TEM (HAADF-STEM) measurements. For $Mo_2C$/NG, Fig. 2a, b and Supplementary Fig. 2 show that high-density and discrete $Mo_2C$ NPs (diameter <5 nm) are uniformly loaded on NG layers. In contrast, in H-$Mo_2C$/NG, GB-rich $Mo_2C$ NSs with a uniform thickness of ~1.0 nm, corresponding to 2 ~ 3 layers of unit cells, are well-distributed on NG (Fig. 2c, d and Supplementary Figs. 3, 4), affording a very high proportion of exposed active sites. Procession electron diffraction (PED) measurements under TEM were employed to collect orientation maps and extract GB density. As shown in Fig. 2e and Supplementary Fig. 5, the crystallographic orientations of each grain in the polycrystalline microstructure are random, and especially abundant intersections between domain walls can be observed. The GB density of $Mo_2C$ NSs in $Mo_2C$ hybrids is summarized in Supplementary Fig. 6, where the statistical region of each sample is ~ 20,000 $nm^2$. Obviously, H-$Mo_2C$/NG possesses the highest GB density of 138 ± 3 $\mu m^{-1}$, clearly demonstrating that the water content in $MoO_3$/RGO effectively manipulates the GB

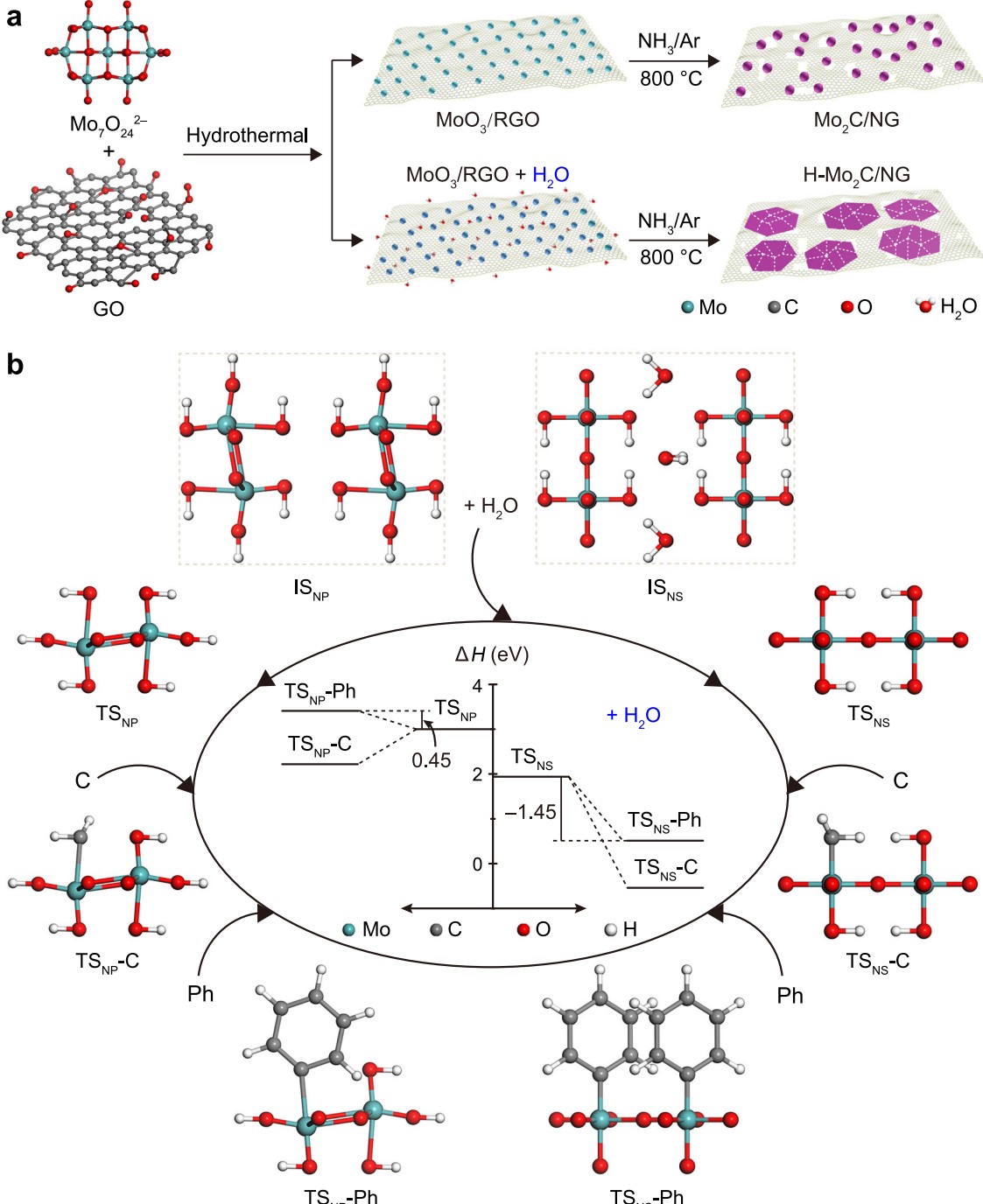

**Fig. 1 | Formation of H-Mo₂C/NG. a** Schematic diagram of the synthesis of Mo₂C/NG and H-Mo₂C/NG. **b** Various reaction states of IS$_{NP}$ and IS$_{NS}$ along the reaction pathway. Inset shows the corresponding reaction $\Delta H$ diagram. The reaction $\Delta H$ for TS$_{NS}$ conversion to TS$_{NS}$-C (−2.48 eV) is lower than that of TS$_{NP}$ conversion to TS$_{NP}$-C (−0.73 eV), indicating that water promotes the formation of Mo₂C during carbonization process.

density of Mo₂C NSs, consistent with TEM results (Supplementary Figs. 7–9 and Supplementary Note 1). It is worth mentioning that the supply of NH₃ (Supplementary Fig. 10 and Supplementary Note 2) and the sufficient oxygen-containing functional groups on GO surface (Supplementary Figs. 11–14 and Supplementary Note 3) are necessary for the formation of GB-rich Mo₂C NSs. Furthermore, tens of HAADF-STEM images clearly reveal that Mo₂C NSs composed of individual grains with nanometer size (Fig. 2f and Supplementary Fig. 15), which forms chemically connected interfaces with numerous GBs and triple junctions, indicating full exposure of high-density active sites in electrocatalysis.

## Atomic structure of GBs on Mo₂C NSs in H-Mo₂C/NG

We employed HAADF-STEM characterization to examine the atomic structure of GBs. The inspection of dozens of boundary locations systematically found that the nanocrystalline grains in Mo₂C NSs are seamlessly stitched with irregular GBs (Fig. 3 and Supplementary Fig. 16). Figures 3a, a1–a3 obviously display that two Mo₂C nanograins form a corrugated GB with an angle of 15°, as well as stacking faults (SFs) and off-centered Mo columns. Furthermore, a conspicuous strain concentration is discovered in the vicinity of the GB (Fig. 3b). Compressive strain (−0.136 ± 0.036) and tensile strain (0.801 ± 0.484 and 0.107 ± 0.086) coexist (Supplementary Fig. 17), ensuring that the

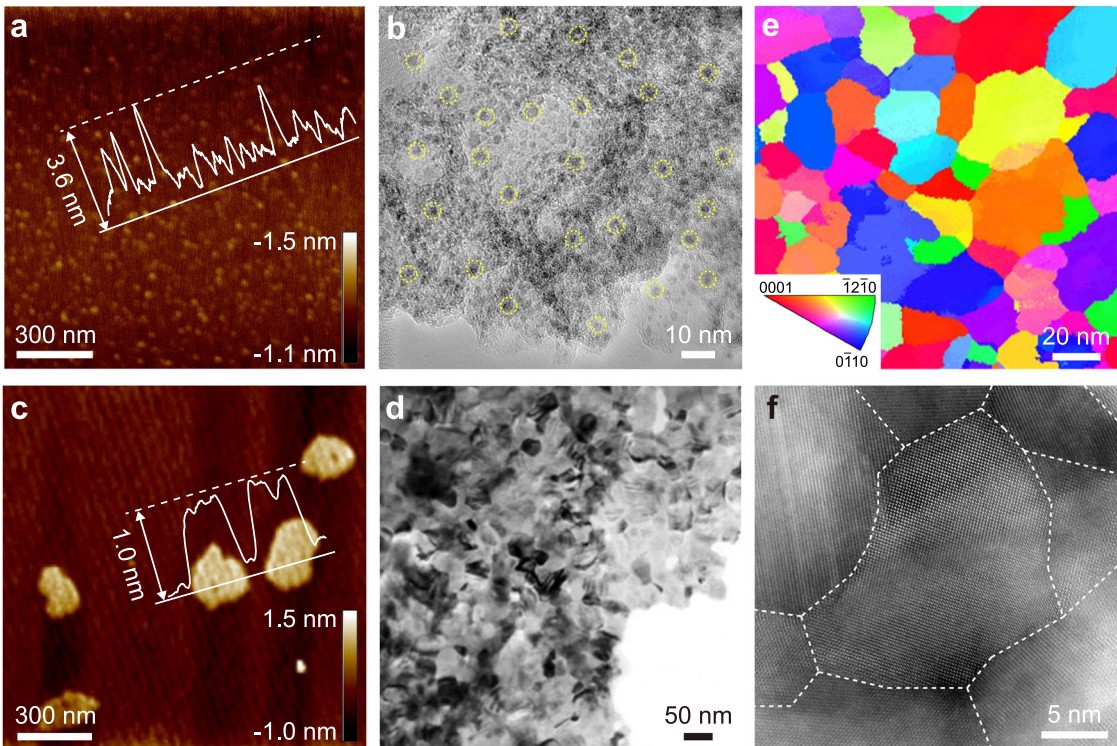

**Fig. 2 | Structural characterization of Mo₂C hybrids. a, c** AFM images of **a** Mo₂C/NG and **c** H-Mo₂C/NG and their corresponding height profiles. **b, d** TEM images of **b** Mo₂C/NG and **d** H-Mo₂C/NG. Mo₂C NPs are highlighted by yellow dotted circles. **e** Inverse pole figure (IPF) images from PED analysis of H-Mo₂C/NG. **f** Representative HADDF-STEM image of H-Mo₂C/NG, the dashed lines point out the irregular GBs.

average strain of Mo₂C NSs almost offsets. In addition, Fig. 3c demonstrates sharp GBs in the same specimen, offering an alternative view. The atomic structure of Mo₂C NS is directly determined by characterizing the nanostructure along two zone axes, where two Z-contrast images closely resemble the structural projection of hexagonal close-packed (hcp) Mo₂C, and their fast Fourier transform (FFT) patterns can be well indexed by hcp cells along their $[1\bar{2}1\bar{3}]_h$ and $[01\bar{1}1]_h$ directions, respectively (Fig. 3c1, c2). Where misorientation angle (38°), step dislocations, and terraced step edges are observed (Fig. 3c3, c4). Particularly, the misorientation angle distribution of observed adjacent grains reveals that the misorientation angle of GBs mostly ranges from 18° to 39° (Supplementary Fig. 18). High-angle GBs are the dominant type of planar defects in all of the Mo₂C NSs examined, distinguishing from conventional Mo₂C nanocrystals grown on graphene without water involved, where low-angle GBs predominate[18]. Compared with low-angle GBs, high-angle irregular GBs have a greater degree of atomic disorder, which generates a suitable electronic structure[19], so as to optimize the H adsorption behavior. Impressively, Mo₂C NSs also occur a topotactic transition from a conventional phase of hcp to a metastable phase of face-centred cubic (fcc), which highlights the controlled synthesis of Mo₂C NSs with unconventional crystal phases through water-assisted carbonization process. Figure 3d, e and Supplementary Fig. 19 show representative HAADF-STEM images of fcc/hcp GB superstructures. Along the fcc/hcp heterophase edges, both hcp phase with "ABAB" stacking and fcc phase with "ABCABC" stacking, with the coexistence of twin boundaries (TBs) and SFs are observed along the close-packed $[110]_f/[11\bar{2}0]_h$ direction. This fcc/hcp heterophase GBs regulate the atomic arrangement of Mo₂C lattice, which modifies the electronic structure of Mo₂C hybrid, thus enhancing the efficiency of hydrogen production[20]. In fact, the fcc and hcp phases of Mo₂C nanocrystals are strongly controlled by carbon activity[21]. Meanwhile, the reaction $\Delta H$ calculation clearly reveal that in anhydrous system, only the C atoms from graphene edges participate the formation of hcp Mo₂C; while in aqueous system, carbon sources

from edge and basal plane of the graphene lattice are both active and readily react with the unique Mo−O configuration to produce Mo₂C NSs with a mixed fcc and hcp phases (Fig. 1b). On the basis of the aforementioned characterizations, the crystalline structure of Mo₂C NSs can be schematically illustrated in Fig. 3f, where the hcp/hcp and fcc/hcp GBs appear alternately. As a result, atomic structure analyzes clarify the atomic configuration at GBs in Mo₂C NSs and reveal the obvious lattice distortion, accompanied by abundant atomic step edges, dislocations, TBs, and SFs, which can effectively alter the surface electronic structure.

Crystal and surface electronic structure of Mo₂C hybrids were investigated by X-ray diffraction (XRD) and X-ray photoelectron spectroscopy (XPS) measurements. The diffraction patterns of H-Mo₂C/NG and Mo₂C/NG in Fig. 4a match with the hcp Mo₂C (PDF#35-0787), which confirms the identity of Mo₂C. Compared to Mo₂C/NG, the diffraction peak intensity of H-Mo₂C/NG is significantly enhanced with a narrower full width at half maxima, indicating that the introduction of water enables the epitaxial growth of Mo₂C grains to harvest large-sized NSs[22]. No diffraction peaks of fcc Mo₂C are detected in Mo₂C hybrids (Supplementary Fig. 20), which could be attributed to the strong destruction of the periodic structure for the fcc lattice because of the alternating arrangement of fcc and hcp Mo₂C lattices[22]. Moreover, the Mo 3d spectra of H-Mo₂C/NG and Mo₂C/NG are deconvoluted into four pairs of peaks (Fig. 4b), indicating that four oxidation states for molybdenum species (+2, +3, +4, and +6) exist in Mo₂C hybrids. Compared to Mo₂C/NG, a negative shift of 0.17 eV is observed on Mo²⁺ in H-Mo₂C/NG, indicating the electron accumulation of Mo atoms near GBs in Mo₂C NSs, which is beneficial to weaken the binding strength of Mo−H and thus enhance the HER activity[6]. Meanwhile, the Mo−C resonance in C K-edge X-ray absorption near-edge structure (XANES) spectra for H-Mo₂C/NG shifts to a higher photon energy zone compared to that of Mo₂C/NG (Fig. 4c), which could be attributed to the enhanced orbital hybridization between the C and Mo elements, that caused by lattice distortion near GBs, facilitating

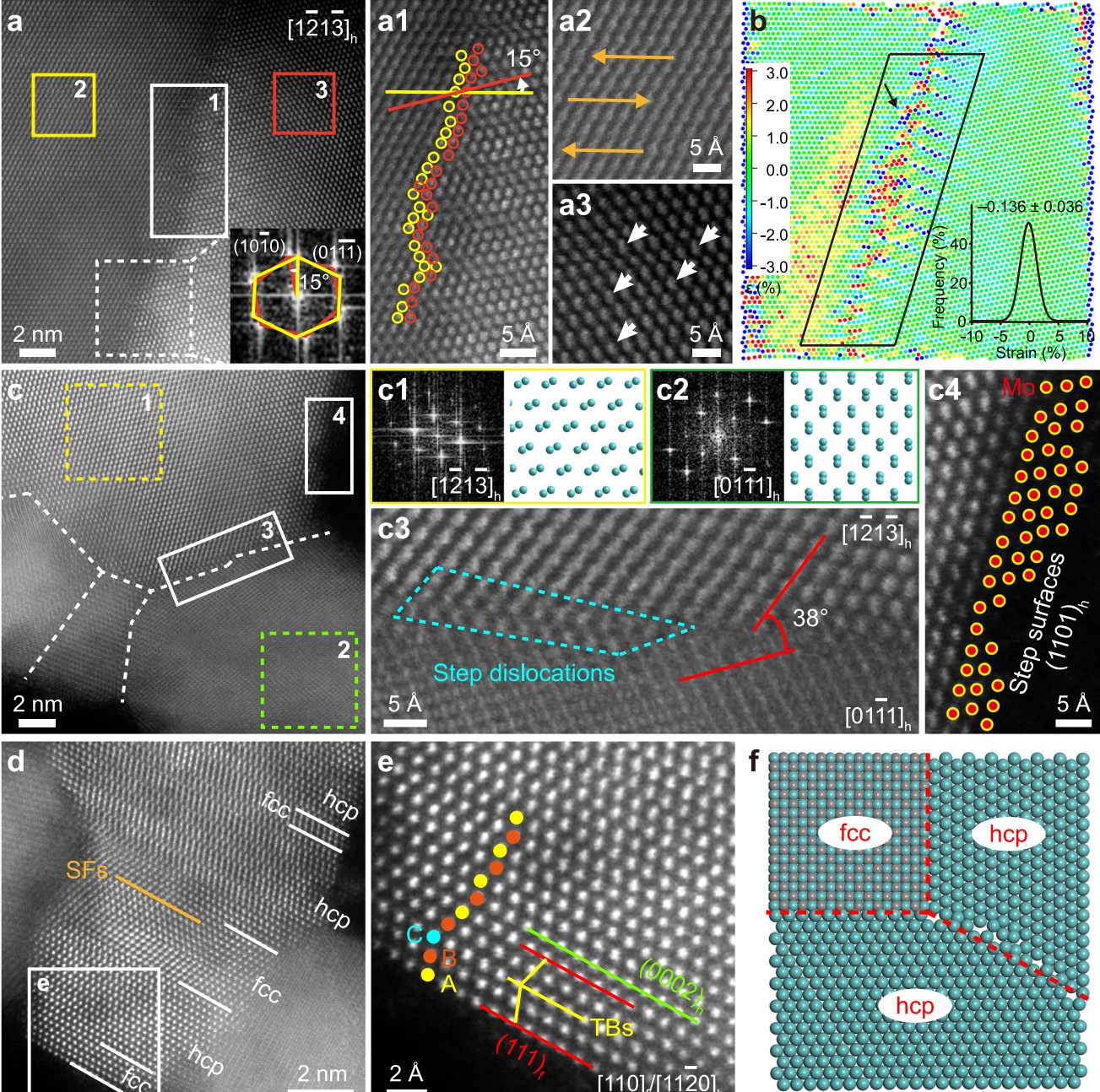

**Fig. 3 | Local atomic structure of GBs on $Mo_2C$ NSs in $H-Mo_2C/NG$. a** A tilted GB in $Mo_2C$ NS. Inset of the FFT pattern indicates a 15° misorientation between grains. **a1** Enlarged HAADF-STEM image of region 1 in **a**. Yellow and tangerine circles represent the Mo atoms on the left and right sides of GB, respectively. **a2, a3** Enlarged HAADF-STEM images of regions 2 and 3 in **a**. The orange and white arrows represent SFs and displacements, respectively. **b** Lattice shear distortion determined from panel **a**, analyzed using the lattice distortion analysis (LADIA) programme. Inset is the quantitative distribution of shear strain for atoms in black-boxed region. The arrow indicates the direction of strain. The strains near GBs range from −5% to 5%. The atoms are colored according to strain scale, that is, colors for positive values represent tensile strain and those for negative values represent compressive strain. **c** The hcp/hcp lattices along the $[1\bar{2}1\bar{3}]_h/[01\bar{1}\bar{1}]_h$ direction, the white dashed lines point out the irregular GBs. **c1, c2** FFT patterns taken from regions 1 and 2 in **c**, and the corresponding projected supercell models of hcp $Mo_2C$. The cyan balls

represent Mo atoms. **c3, c4** Enlarged HAADF-STEM images of regions 3 and 4 in **c** displaying tilt GBs, step dislocations, and stepped surfaces. Step dislocations, caused by the insertion of additional planes into the otherwise self-contained lattice, highlighted with the cyan dotted box. The atomic displacement arose from this distortion at GBs is usually much greater than the relaxation displacement occurring on the crystal surface as well as the displacement caused by the coherent strain through epitaxial thin-film growth[18]. **d** The fcc/hcp lattices along the $[110]_f/[11\bar{2}0]_h$ direction. The fcc/hcp GBs are marked with white solid lines. **e** Enlarged HAADF-STEM image of the square region in **d**. Yellow, tangerine, and cyan dots, respectively, represent planes A, B, and C. Typical defects, including SFs and TBs are marked with orange and yellow lines, respectively. **f** Schematic illustration of hcp/hcp and fcc/hcp GBs in $Mo_2C$ NSs, the red dashed lines point out the irregular GBs.

electron transfer[23]. In addition, to further understand the role of water during carbonization process, a series of $Mo_2C$ hybrids synthesized with various water contents were also subjected to XPS measurement. As water content increases, the peak intensity of Mo−C

(Supplementary Fig. 21) and Mo−N (Supplementary Fig. 22) species increases slightly, indicating that water promotes the formation of nitrogen-doped $Mo_2C$. As revealed by the theoretical calculation that the reaction $\Delta H$ of $TS_{NS}$ to produce $TS_{NS}$-N (−1.05 eV) is lower than that

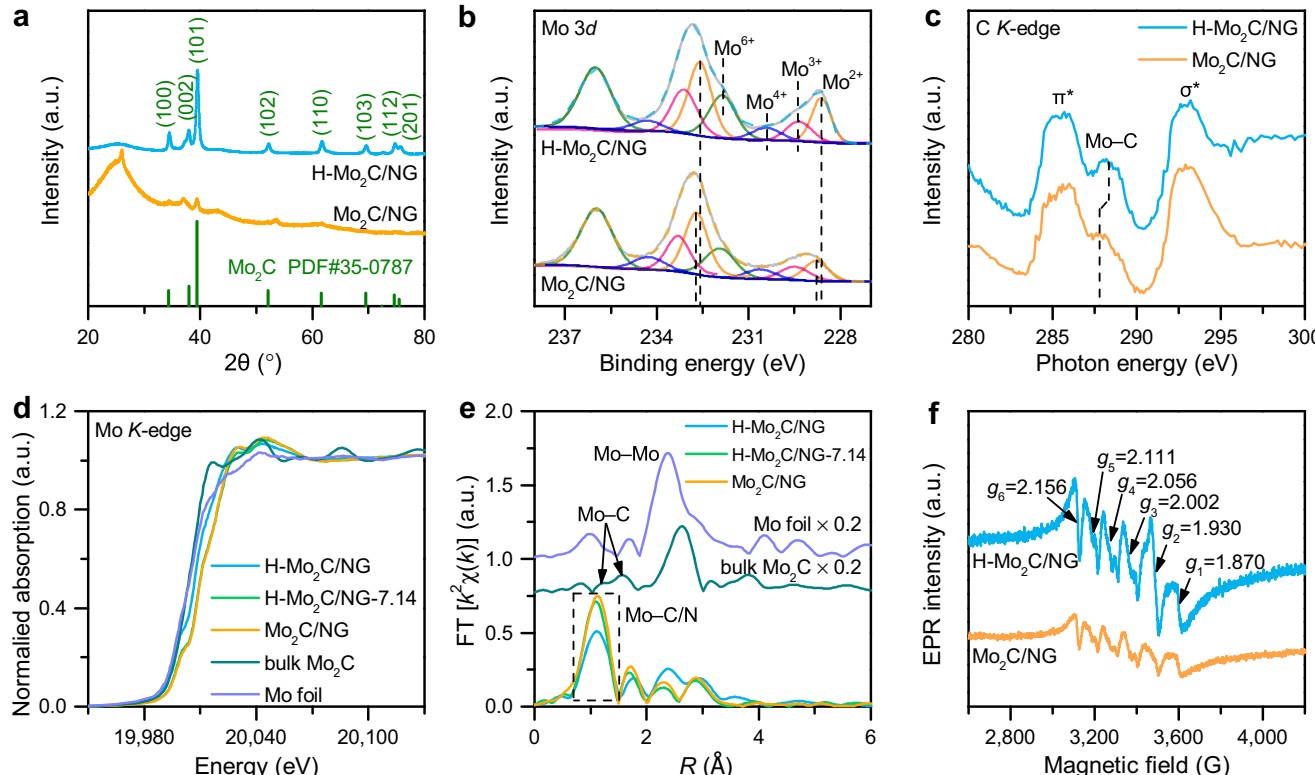

**Fig. 4 | Spectroscopy characterization of Mo₂C hybrids. a** XRD patterns of H-Mo₂C/NG and Mo₂C/NG. The peaks near 2θ = 26° and 43° are assigned to the (002) and (100) reflections of graphene, respectively, and the weak peaks at 37° and 53° are considered to be the ($\overline{2}$11) and ($\overline{2}$22) planes of MoO₂ (PDF#32-0671), respectively. **b** The high-resolution Mo 3*d* XPS spectra of H-Mo₂C/NG and Mo₂C/NG. Mo⁴⁺ and Mo⁶⁺ can be attributed to MoO₂ and MoO₃ caused by the oxidation of molybdenum species in ambient atmosphere, respectively. The Mo²⁺ and Mo³⁺ are assigned to carbides and nitrides, respectively, which are known to serve as active sites for HER[6]. **c** C K-edge XANES spectra of H-Mo₂C/NG and Mo₂C/NG. The

characteristic resonances of C = C π* and C–C σ* originating from NG support. **d** Mo K-edge XANES spectra of H-Mo₂C/NG, H-Mo₂C/NG-7.14, Mo₂C/NG, and the reference samples. H-Mo₂C/NG-7.14 and Mo₂C/NG have a higher half-edge energy than that of bulk Mo₂C reference, which arises from the charge-transfer from Mo to C atoms on the NG-supported Mo₂C[8]. **e** Mo K-edge Fourier transform EXAFS spectra of H-Mo₂C/NG, H-Mo₂C/NG-7.14, Mo₂C/NG, and the reference samples. **f** EPR spectra of H-Mo₂C/NG and Mo₂C/NG. H-Mo₂C/NG shows the EPR signals at $g_2$ = 1.930 (Mo–N species[36]), $g_1$ = 1.870 and $g_4$ = 2 .056 (Mo⁵⁺ species[37]), followed by $g_5$ = 2.111 and $g_6$ = 2.156 (the resonance on conduction electrons[38]).

of TS$_{NP}$ to produce TS$_{NP}$-N (−0.31 eV) (Supplementary Fig. 23), indicating that unique Mo–O configuration induced by water prefers to bind with N atoms forming Mo–N species.

To precisely probe the local coordination structure of GBs in Mo₂C nanosheets, XANES and extended X-ray absorption fine structure (EXAFS) at Mo K-edge were conducted. Compared to H-Mo₂C/NG-7.14 and Mo₂C/NG, the pre-edge of H-Mo₂C/NG obviously shifts to lower energy (Fig. 4d), indicating the increased average electron density around Mo atoms, consistent with XPS results. The details of Mo local environment can be described through the $k^2$-weighted Fourier-transformed EXAFS (FT-EXAFS) in R-space. Mo₂C hybrids display a prominent peak at 1.30 Å originated from the Mo–C/N vector; with the increase of water content, the peak intensity of Mo–C/N path gradually decreases, indicating a low coordination number (CN) for Mo atoms in Mo₂C (Fig. 4e)[24]. In addition, comparing with Mo foil and bulk Mo₂C, Mo₂C hybrids exhibit relatively weak Mo–Mo vectors at 2.30 and 2.90 Å; these Mo–Mo paths in H-Mo₂C/NG (2.38 and 2.95 Å) are slightly stretched compared to H-Mo₂C/NG-7.14 (2.29 and 2.87 Å) and Mo₂C/NG (2.29 and 2.87 Å). These results indicate that numerous Mo atoms in H-Mo₂C/NG are misplaced and deviated from the normal crystalline position, which could be attributed to the high-angle GBs with large lattice strain in Mo₂C NSs[25]. Furthermore, quantitative EXAFS fitting was performed using DFT optimized hcp Mo₂C structure to extract the structural parameters, and the fitting results are displayed in Supplementary Fig. 24. Briefly, compared to H-Mo₂C/NG-7.14 and Mo₂C/NG, H-Mo₂C/NG has two different Mo–C/N paths and low CN (Supplementary Table 1), highlighting the structural distortion and

carbon defects, which further confirms the construction of GBs in Mo₂C NSs. Moreover, the surface defects in Mo₂C NSs are further resolved with the electron paramagnetic resonance (EPR) spectroscopy. As illustrated in Fig. 4f, the EPR signal at $g_3$ = 2.002 for H-Mo₂C/NG is attributed to the unpaired electrons trapped on carbon defects at Mo₂C GBs;[26] where a slightly lower signal is observed for Mo₂C/NG, demonstrating relatively high concentration of coordination unsaturated Mo atoms in Mo₂C NSs, consistent with EXAFS results. The above results demonstrate that the water-assisted carbothermal reaction strategy successfully constructs ultrathin Mo₂C NSs with high-density GBs anchored on NG support, which provides sufficient active sites and the possibility of modulating the interface atomic and electronic configurations.

## Electrocatalytic HER performance

The catalytic performance of Mo₂C hybrids was investigated by a three-electrode system in 0.5 M H₂SO₄ and 1.0 M KOH solutions (see more details in Methods). As revealed in Fig. 5a, H-Mo₂C/NG shows the highest HER performance among all catalysts in acidic media, and only requires an overpotential of 10 mV to achieve the geometric current density of −10 mA cm⁻² ($\eta_{10}$), significantly superior to the Mo₂C/NG ($\eta_{10}$ = 138 mV) and even Pt/C catalyst ($\eta_{10}$ = 19 mV). Furthermore, the Tafel slope of H-Mo₂C/NG is 38 mV dec⁻¹ (Fig. 5b), which far surpasses that of Mo₂C/NG (137 mV dec⁻¹) and is even comparable to that of state-of-the-art Pt/C catalyst (34 mV dec⁻¹), suggesting that H-Mo₂C/NG possesses more fast kinetics with the Heyrovsky reaction as the rate-determining step[18]. The exchange current density ($j_0$) of H-Mo₂C/NG is

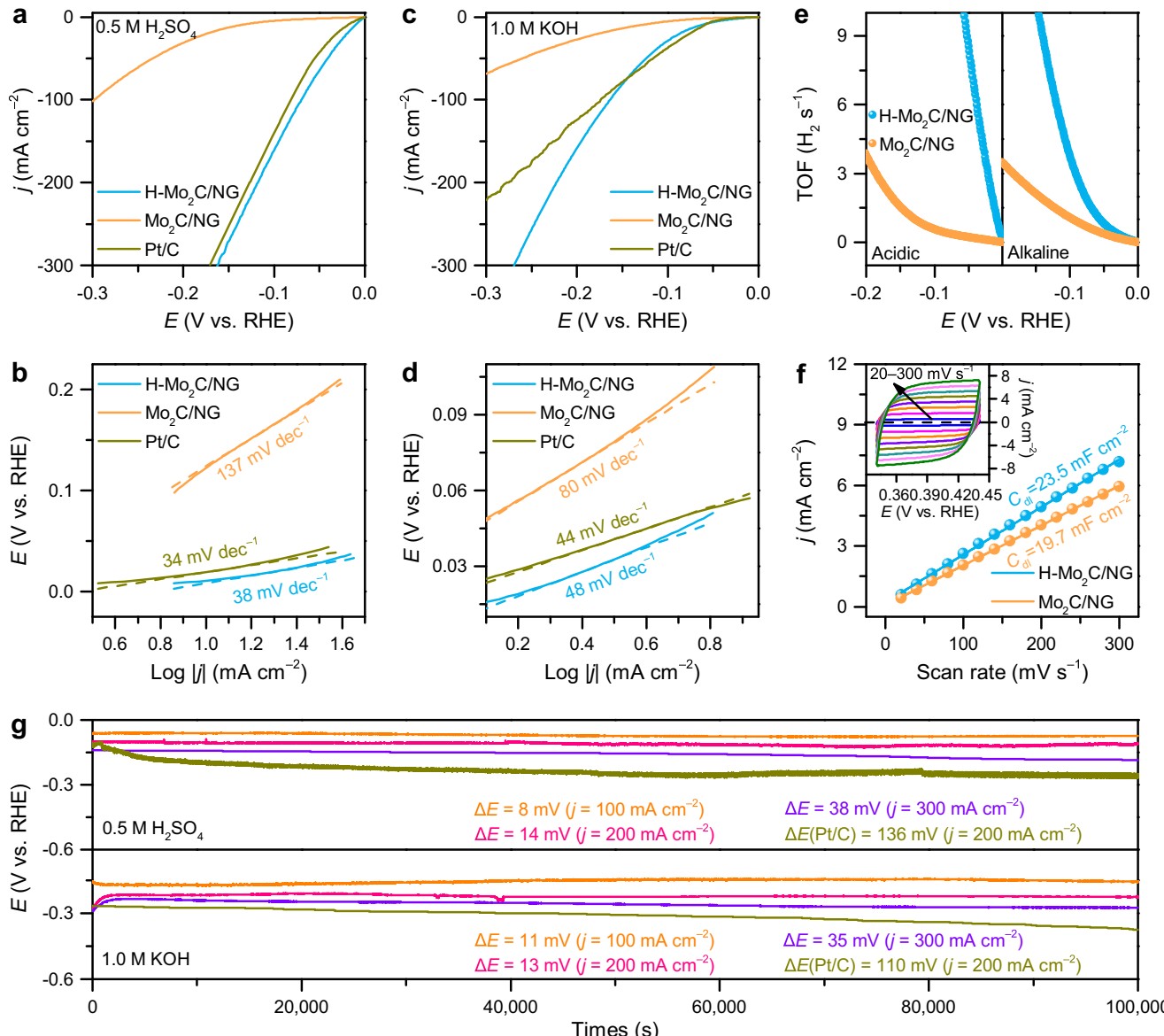

**Fig. 5 | HER performance. a, c** The polarization curves and **b, d** the relevant Tafel plots of H-Mo₂C/NG, Mo₂C/NG, and Pt/C in **a, b** 0.5 M H₂SO₄ and **c, d** 1.0 M KOH, respectively. **e** TOF values of H-Mo₂C/NG and Mo₂C/NG in 0.5 M H₂SO₄ and 1.0 M KOH. **f** The plots of the current density versus the scan rate for H-Mo₂C/NG and Mo₂C/NG in 1.0 M KOH. Inset is cyclic voltammetry (CV) cycles of H-Mo₂C/NG. **g** Galvanostatic measurement of H-Mo₂C/NG in 0.5 M H₂SO₄ and 1.0 M KOH at $j = -100$, $-200$, and $-300$ mA cm⁻², respectively.

calculated to be 5.47 mA cm⁻², which is much higher than that of Mo₂C/NG (0.80 mA cm⁻²) (Supplementary Table 2), further underscoring favorable HER kinetics. Meanwhile, H-Mo₂C/NG also demonstrates promising HER activity in alkaline media. H-Mo₂C/NG merely requires overpotentials of 63 and 163 mV to reach −10 and −100 mA cm⁻², respectively, which are comparable to commercial Pt/C catalyst ($\eta_{10}$: 57 mV and $\eta_{100}$: 173 mV) and significantly better than those of Mo₂C/NG ($\eta_{10}$: 107 mV and $\eta_{100}$: 388 mV) (Fig. 5c). Compared to Mo₂C/NG catalyst, H-Mo₂C/NG also affords a lower Tafel slope of 48 mV dec⁻¹ (Fig. 5d) and higher $j_0$ value (0.75 mA cm⁻²), which reveals the same favorable HER kinetics as in acidic media for H-Mo₂C/NG. To further study the effect of GBs on HER activity, we performed electrocatalytic hydrogen evolution tests on a series of Mo₂C hybrids. Obviously, as the water content increases, the $\eta_{10}$ and Tafel slopes on Mo₂C hybrids firstly decrease and then increase (Supplementary Figs. 25-28), where H-Mo₂C/NG with the highest GB density achieves the best HER performance. These results suggest that the GBs in Mo₂C NSs can effectively enhance the intrinsic HER activity of the Mo₂C hybrids.

Additionally, to gain insight into the superior catalytic activity of H-Mo₂C/NG, the turnover frequency (TOF) was calculated (Supplementary Fig. 29 and Supplementary Note 4). As displayed in Fig. 5e, H-Mo₂C/NG affords the TOF values of 21.33 and 3.81 H₂ s⁻¹ at $\eta = 100$ mV in acidic and alkaline electrolytes, respectively, which are quite larger than those of Mo₂C/NG (0.57 and 1.08 H₂ s⁻¹), as well as other control samples (Supplementary Fig. 30), confirming the high HER activity of H-Mo₂C/NG. To further identify the intrinsic activity of H-Mo₂C/NG, geometric current density ($j$) was normalized by electrochemical active surface area (ECSA; $j_{ECSA}$), which was determined by deriving the electrochemical double-layer capacitance ($C_{dl}$). The $C_{dl}$ of H-Mo₂C/NG (23.8 mF cm⁻²) is slightly higher than that of Mo₂C/NG (19.7 mF cm⁻²), suggesting the close ECSA values (Fig. 5f and Supplementary Note 5). Yet, H-Mo₂C/NG delivers the $j_{ECSA}$ values of 109.53 and 10.57 μA cm⁻² at $\eta = 50$ mV in 0.5 M H₂SO₄ and 1.0 M KOH (Supplementary Figs. 31, 32), which are evidently higher than those of Mo₂C/NG (3.70 and 6.23 μA cm⁻²) and Pt/C (80.16 and 9.86 μA cm⁻²), respectively, revealing better intrinsic HER activity of H-Mo₂C/NG with

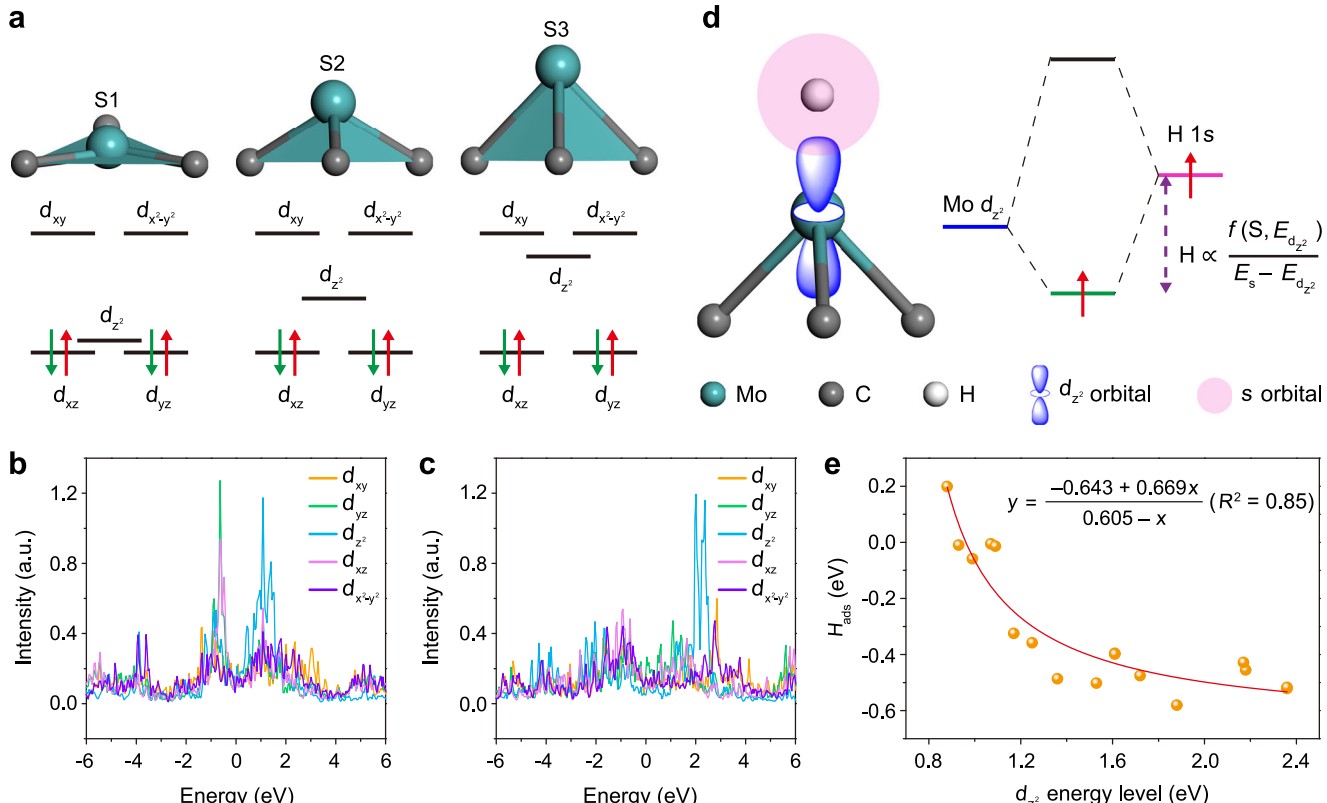

**Fig. 6 | Relationship between HER activity and the $d_z^2$ orbital energy level.** **a** Structural models and relevant electronic states of Mo $d$ orbital for $MoC_3$ configurations. S1: $MoC_3$ triangular plane, S2: intermediate height, S3: high height. PDOS of Mo $d$ orbital for **b** S2 and **c** S3 structures. **d** Schematic diagram of hybridization of H $1s$ orbital and Mo $d_z^2$ orbital. **e** Correlation of the Mo $d_z^2$ orbital energy level and $H_{ads}$, inset is the corresponding fitting equation.

massive GBs, even outperforming benchmark Pt/C catalyst. In comparison with non-precious metal HER electrocatalysts reported recently (Supplementary Tables 3, 4), H-Mo₂C/NG exhibits almost the lowest overpotential and Tafel slope, illustrating the super HER activity in pH-universal electrolyte. Besides, H-Mo₂C/NG has a small charge-transfer resistance ($R_{ct}$) (2.5 and 2.1 Ω) in 0.5 M H₂SO₄ and 1.0 M KOH, respectively, which is much lower than those of Mo₂C/NG (5.5 and 12.5 Ω) and other control samples (Supplementary Fig. 33). This result indicates that H-Mo₂C/NG possesses much fast electron transfer kinetics during HER process, which is attributed to the unique structure of Mo₂C NSs with chemically connected GBs, shortening the electron transfer pathway between active sites[27].

For real applications, the stability and durability of H-Mo₂C/NG were further analyzed. As illustrated in Supplementary Fig. 34, after 10,000 or even 50,000 cycles in 0.5 M H₂SO₄ and 1.0 M KOH, the shifts of polarization curves for H-Mo₂C/NG are negligible compared with the initial curves. Meanwhile, we performed aggressive long-term stability examine on H-Mo₂C/NG by continuous galvanostatic measurement (Fig. 5g). After 100,000 s of operation at $j = -100$, $-200$, and $-300$ mA cm⁻², the potentials of H-Mo₂C/NG only increase by 8, 14, and 38 mV in 0.5 M H₂SO₄, and 11, 13, and 36 mV in 1.0 M KOH, respectively. In contrast, the potentials of Pt/C catalyst at $j = -200$ mA cm⁻² increase by 136 and 110 mV in 0.5 M H₂SO₄ and 1.0 M KOH, respectively. In addition, TEM characterization shows that Mo₂C nanocrystals in H-Mo₂C/NG still maintain an ultrathin nanosheet structure with highly exposed high-density GBs after electrochemical cycling (Supplementary Fig. 35). XRD characterization displays that Mo₂C is well crystallized, and no other phase is experimentally detected (Supplementary Fig. 36). The chemical valence state of Mo species remains almost unchanged, further indicating the robustness of H-Mo₂C/NG in the electrocatalytic HER process (Supplementary Fig. 37). It should be noted that most synthesized 2D Mo₂C catalysts cannot withstand a

high current density and operate for a long-lasting lifetime in a wide pH range because of the weak interfacial adhesion with adjacent nanocrystals or substrates[11,13]. In this work, the excellent physicochemical stability of H-Mo₂C/NG can be attributed to the chemical connection of Mo₂C GBs and strong interaction of Mo₂C NSs with NG sheets through Mo−C bonds. Moreover, the general synthetic strategy of GB-rich NSs can also be extended to other transition metal carbide systems (Supplementary Fig. 38), such as vanadium (Supplementary Fig. 39), niobium (Supplementary Fig. 40), tantalum (Supplementary Fig. 41), and tungsten (Supplementary Fig. 42), and yield excellent HER performances in pH-universal electrolyte (Supplementary Fig. 43), demonstrating the generality of this method, which provides an alternative way to develop high-efficiency catalysts through GB engineering.

## DFT calculations

To understand the correlation between the electronic structure and catalytic performance in GB-rich Mo₂C plane, DFT calculations were conducted. According to the HAADF-STEM images of Mo₂C NSs (Fig. 3 and Supplementary Figs. 16, 19), the computational models of fcc/hcp, fcc/fcc, and hcp/hcp GBs were constructed, respectively. The DFT calculation shows that the fcc/hcp Mo₂C slab surface exists various kinds of trigonal pyramid-like structures (Supplementary Fig. 44), in which Mo atom locates at top and pyramidally coordinates with three C atoms at bottom, forming a typical trigonal pyramid ($MoC_3$) configuration (Fig. 6a). Depending on the height of the pyramid-shaped $MoC_3$ skeleton on the fcc/hcp Mo₂C superlattice surface, the corresponding configuration of $MoC_3$ can be schematically classified into three types: (i) approximative triangular plane (S1), (ii) intermediate height (S2), and (iii) high height (S3). The hcp Mo₂C surface consists of S1 and S3 structures, in which the S3 structure typically represents common bulk Mo₂C configuration and is predominant; on the fcc Mo₂C surface, $MoC_3$ pyramids only present S3 structure. Notably, the

fcc region of the fcc/hcp GB is contracted interface with a smaller Mo–Mo distance corresponding to the S3 structure. While the hcp region of the fcc/hcp GB exhibits an elongated interface with a larger Mo–Mo distance, where $MoC_3$ pyramids at GB have a medium height and belong to S2 structure.

Furthermore, to find the rule for the GB effect on HER electrocatalysis, the crystal field analyzes of $MoC_3$ pyramids for S2 and S3 structures were performed with the projected density of state (PDOS) calculation, as illustrated in Fig. 6b and c, respectively. Evidently, with the increase of the height (from bottom $C_3$ plane to atop Mo atom), the energy level of the Mo $d_z^2$ orbital also increases correspondingly. Consequently, these three low-lying orbitals further undergo symmetry breaking, rising the Mo $d_z^2$ orbital state into distinct energy levels[28] (Fig. 6a). In the electrocatalytic HER process, the adsorption of H atoms on the Mo atoms forms σ bonds, which originate from the orbital hybridization between the H $1s$ orbitals and the Mo $d_z^2$ orbitals (Fig. 6d). According to the Sabatier principle, the ideal electrocatalyst should bind H intermediate neither too strongly nor too weakly[29]. Therefore, the optimized adsorption free energy and fast HER kinetics can be achieved by manipulating the bond strength between the H $1s$ orbital and Mo $d_z^2$ orbital[4].

The bond strength is defined as:

$$E \propto \frac{f(S, E_{d_{z^2}})}{E_s - E_{d_{z^2}}} \qquad (1)$$

Where $E_s$ and $E_{d_{z^2}}$ are the energy levels of H $1s$ orbital and Mo $d_z^2$ orbital, respectively. S is orbit overlap population of H $1s$ orbital and Mo $d_z^2$ orbital. Within these contexts, we calculated the H adsorption free energy ($H_{ads}$) of surface Mo atoms on the model of fcc/hcp GB, and studied its relationship with the Mo $d_z^2$ orbital energy level (Fig. 6e). Where the numerical fitting over the calculated $H_{ads}$ and the Mo $d_z^2$ orbital energy level follows the similar function as the Eq. (1) of bond strength, that is, the adsorption energy varies with the Mo $d_z^2$ orbital energy level. As a result, the intrinsic HER activity exhibits a strong dependence on the Mo $d_z^2$ orbital energy level, which rationalizes the role of $d_z^2$ orbital energy level as one of the most important activity descriptors for HER in $Mo_2C$ systems. Compared with the state-of-the-art descriptors such as band-energy theory and work function that describe the properties of the whole surface[30], the $d_z^2$ orbital energy level is more relevant to the active site of the catalysts. Therefore, the descriptor $d_z^2$ orbital energy level could be used as a figure of merit for designing transition metal catalyst with well-defined active sites for electrocatalytic HER reaction. Additionally, the $H_{ads}$ on the Mo atoms near the fcc/fcc and hcp/hcp homophase $Mo_2C$ GBs also display close-to-zero values, such as −0.044, 0.035, and 0.038 eV (Supplementary Figs. 45 and 46), which also contribute to the excellent HER activity. As a result, the DFT results confirm that GBs in ultrathin $Mo_2C$ NSs can significantly alter the electronic structures of H-$Mo_2C$/NG electrocatalyst, and the intrinsic HER activity strongly depends on the Mo $d_z^2$ orbital energy level controlled by the $MoC_3$ pyramid configuration.

## Discussion

In summary, ultrathin $Mo_2C$ NSs with rich GBs supported on NG have been developed by facile hydrothermal and water-assisted carbothermal reactions with GO as carbon source. During carbonization process, water induces structural evolution of $Mo_2C$ nanocrystals from NPs to NSs and controls the GB density of $Mo_2C$ NSs as well. In H-$Mo_2C$/NG, the dense polycrystalline $Mo_2C$ NSs with massive irregular GBs are chemically connected to NG through Mo–C bonds, which afford ultrahigh fraction of active sites for electrolysis, exhibiting high reproducibility in terms of structure and performance. The inherent HER activity of H-$Mo_2C$/NG electrocatalyst strongly depends on the $d_z^2$ orbital energy level of Mo atoms, which is controlled by the $MoC_3$ pyramid configuration. This work provides a horizon for rationally designing GB interfaces of transition metal electrocatalysts to achieve excellent HER activities, and establishes a descriptor of the $d_z^2$ orbital energy level for the design of high-performance catalysts.

## Methods

### Chemicals

All chemicals were purchased from commercial sources (Sigma Aldrich, Energy Chemical) without further purification. GO was synthesized from graphite flakes by the improved Hummers method[31].

### Synthesis of H-$Mo_2C$/NG

First, the GO suspension was prepared by adding 0.16 g GO into 80 mL deionized water and sonicating for 10 h. Typically, 0.4470 g $(NH_4)_6Mo_7O_{24}\cdot4H_2O$ were added into the GO suspension and sonicated for another 4 h. Subsequently, above mixture was transferred to a Teflon-lined autoclave and heated at 190 °C for 12 h to harvest $MoO_3$/RGO intermediate. Next, the $MoO_3$/RGO intermediate was freeze-dried to obtain sponge columns with various water contents, including 0, 0.41, 7.14, 13.52, 18.56, 30.98, and 43.28 wt%. The $MoO_3$/RGO with various water contents were put into a tubular quartz furnace for heat treatment. After pumping and purging the system with Ar for 30 min, the temperature was ramped at 10 °C $min^{-1}$ up to 800 °C with the feeding of Ar (150 sccm) and $NH_3$ (100 sccm). The reaction was allowed to proceed for 3 h and the final product was fast cooled to room temperature by quickly removing the sample from the hot zone of the furnace under the protection of flowing Ar. The final catalysts were named H-$Mo_2C$/NG-X, where X was the water content retained in $MoO_3$/RGO intermediate, for example, H-$Mo_2C$/NG-7.14 (the water content was 7.14 wt%). In our report, H-$Mo_2C$/NG-18.56 and H-$Mo_2C$/NG-0 were denoted as H-$Mo_2C$/NG and $Mo_2C$/NG unless otherwise specified, respectively.

### Material characterization

Morphological structure characterizations of $Mo_2C$ hybrids were performed on SEM (JEOL-JSM-7001F) and TEM (FEI Talos F200X, FEI Titan ChemiSTEM, FEI Titan Themis). HAADF-STEM images were carried out by a Cs-corrected FEI Titan G2 60-300 equipped with a Super-X EDS detector. The LADIA package was used to analysis the strain states of $Mo_2C$ nanocrystals during the loading and unloading of tensile stress[32]. The surface electronic structure was checked by XPS spectra (XPX, PHI-5702). The C and N K-edge NEXAFS spectra were measured at the photoemission end-station at beamline BL10B in the National Synchrotron Radiation Laboratory (NSRL) in Hefei, China. X-ray absorption spectra (XANES and EXAFS) of Mo K-edge were measured on VESPERS beamline at the Canadian Light Source, by scanning a double crystal Si (111) monochromator and collecting emitted X-ray fluorescence.

### Electrochemical measurements

Electrochemical measurements were conducted on a typical three-electrode system (CH Instruments 760E). All potentials were referenced to a reversible hydrogen electrode (RHE) and without iR compensation. The binder solution was prepared by mixing of 5 wt% Nafion solution (40 μL) with 1 mL of 4:1 v/v deionized water/ethanol. The catalyst ink was then prepared by dispersion of 2 mg $Mo_2C$ hybrids (H-$Mo_2C$/NG, $Mo_2C$/NG, and control samples) into binder solution followed by ultrasonication for 2 h. In $Mo_2C$ hybrids, molybdenum atoms are the active species, and molybdenum loading (~ 8 wt%, Supplementary Table 5) is basically the same. To maintain the same metal load, 8 mg Pt/C catalyst (20 wt% Pt on graphitized carbon, Johnson Matthey) ink was prepared in the same way. The ink (10 μL) was then dripped onto a carbon fiber paper (CFP, 5 mm × 5 mm) as the working electrode. The electrode was allowed to dry at room temperature for at least 24 h before measurement. After drying, a catalyst ($Mo_2C$ hybrids and Pt/C) mass loading of 0.038 mg $cm^{-2}$ was obtained. A graphite rod

as the counter electrode and saturated calomel electrode (SCE) as the reference electrode. All potentials were referenced to RHE: $E_{RHE} = E_{SCE} + (0.242 + 0.059\,pH)$ V. In 0.5 M $H_2SO_4$ and 1.0 M KOH solutions, the polarization curves were obtained at the electrode potential of 0 to 0.45 V using a scan rate of 50 mV s$^{-1}$. The electrochemical EIS was performed from $10^{-2}$ to $10^6$ Hz with an AC voltage of −5 mV. Galvanostatic charge discharge curves of H-Mo$_2$C/NG were recorded at $j = -100$, −200, and −300 mA cm$^{-2}$. Prior to all measurements, the electrochemical system was purged with $H_2$ bubbles for 30 min, and then conducted all electrochemical measurements at room temperature (25 °C) under ambient atmosphere.

## DFT calculations

The first-principles calculations based on the DFT were performed within generalized gradient approximation (GGA). Core electron states were represented by the projector augmented-wave method as implemented in the Vienna ab initio simulation package (VASP)[33,34]. The Perdew–Burke–Ernzerhof exchange-correlation functional and a plane wave representation for the wave function with a cut-off energy of 450 eV were used[35], more details can be found in Supporting Information (Supplementary Note 6). The atomic coordinates of the H adsorption on the fcc/hcp GB model to generate Fig. 6e are listed in VASP CONTCAR format (Supplementary Data 1).

## Data availability

The data supporting the findings of this study are available within the article and its Supplementary Information. Additional data are available from the corresponding authors on reasonable request. Source data are provided with this paper.

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

## Acknowledgements

We thank Dr. B. Chen from Rice University for assistance with XPS spectroscopy. The authors acknowledge X. Guo at Huazhong University of Science and Technology (HUST) and Y. Zhou at Xiamen University for their valuable scientific discussion. The authors acknowledge the National Natural Science Foundation of China (No. 22175109, X.F.), the Natural Science Foundation of Excellent Young Scholars for Shanxi Province (201901D211198, X.F.), the Fund for Shanxi "1331" Project, and Sanjin Scholar for financial support of this research.

## Author contributions

Y.Y., Y.Q., Z.L., and X.F. conceived the experiment and scientific discussions. Y.Y. carried out the syntheses and electrocatalysis measurements. H.L., L.C., X.C., S.W., B.Z., Z.Z., S.C., W.Y., J.D., L.S., W.Z., R.F., J.Z., K.D., and X.-M.Z. carried out the characterizations. Z.L. and X.L. performed the atomic resolution microscopy and PED. Y.Q. carried out the computational investigation and provided the theoretical analysis. Y.Y. and X.F. wrote the paper. All the authors discussed the results and revised the paper.

## Competing interests

The authors declare no competing interests.
