## [Peer Review File · Nature Communications]

Water Induced Ultrathin Mo₂C Nanosheets with High-Density Grain Boundaries for Enhanced Hydrogen EvolutionREVIEWER COMMENTS

Reviewer #1 (Remarks to the Author):

Recommendation: Rejection

Comments:

In the current work, the authors said that their dense grain bounds from Mo₂C nanosheets afford an ultra-high density of active sites for HER. But they did not think about (1) the evolution of nitrogen concentration in N-doped graphene and (2) the effect of nitrogen and oxygen content in Mo₂C. All these can affect the hydrogen evolution reaction in this system. The work did not support the conclusions. There are many flaws in the data analysis, interpretation, and conclusions. This manuscript is prohibited from being published.

1. References are not in the correct positions. Such as reference 9.
2. Should prove the active sites for HER are in the grain boundary from experiments and calculations if their conclusion is correct.

Reviewer #2 (Remarks to the Author):

- What are the noteworthy results?

Good HER performance, simple synthesis methods and reagents. They claim grain boundary control however these results are questionable

- Will the work be of significance to the field and related fields? How does it compare to the established literature? If the work is not original, please provide relevant references.

Many groups have investigated the synthesis of molybdenum nitride and carbide through reaction of moly oxide and ammonia or ammonolysis. In addition, there is a vast field of Mo₂C catalysts using support materials like graphene and rGO. This paper is unique in observing the effects that water content have on the overall reaction and their investigation provide some new insights or understudied to the field. However they also need to include the role that NH₃ plays in the reaction, and if its affects play a role in the formation of nanoparticles, nanosheets, and/or grain boundaries. I feel like the results and discussion in their current form do not justify this as a significant advancement of the field.

- Does the work support the conclusions and claims, or is additional evidence needed?

In short, no. There are some nice results and good data overall, however, I think determining vital information about the grain boundaries and the mechanism of their formation are quite difficult and not necessarily supported by the data. In addition, they should include much more discussion on the amount of Mo-N or Molybdenum nitride present and the role that NH₃ plays in these reactions. Heating MoO₃ in NH₃ likely forms nitride which converts to the carbide in the presence of carbon (rGO). They claim the NH₃ is needed to reduce the rGO and form nitrogen doping, however, they barely mention that there is Mo-N other than the XPS section. If they anneal in absence of NH₃ or use ammonia in the hydrothermal portion of the synthesis instead of the high temperature furnace they can avoid the formation of the nitride and validate that it is simply rGO and Mo₂C not Mo-N responsible for their catalytic results.

- Are there any flaws in the data analysis, interpretation and conclusions? Do these prohibit publication or require revision?

Yes

I don't agree that there is a CVD process going on. It seems like the precursor solid has the carbon and Mo present and then they perform an ammonolysis reaction on the MoO₃ which has been used for decades to make nitrides.

They rarely mention the fact that Mo-nitride is a significant part of the product and do not account for it in the explanation of the results. Nitride is only mentioned once in the document

The role of ammonia should also be described in more detail. Do they get the same number of grain boundaries using inert atmosphere? Does going through a nitride intermediate make a difference on the carbide formation, particle size, grain boundaries etc

They also need to include the nitride and any relevant oxides in the DFT calculations to further explain if they play a role in the catalytic activity and stability of these materials.

They mention the pores of these materials and highlight them in some TEM images but they do not describe the uniformity in size or shape of these. Most of the images simply look like holes left behind from aggregation of smaller particles that leaves voids rather than pore created from a byproduct leaving.

- Is the methodology sound? Does the work meet the expected standards in your field?

Grain boundary analysis is difficult to quantify and prove. Even within the same image (ex supporting figure 6) there are several different particle sizes and grain boundaries present. I have a hard time telling the difference between the different samples. This make the data in supporting figure 8 and the major theme of this paper difficult to prove.

- Is there enough detail provided in the methods for the work to be reproduced?

Yes

Reviewer #3 (Remarks to the Author):

This work by Yang and co-authors reports on electrocatalytic activity of a modified Mo₂C nanosheet for hydrogen evolution reaction (HER). They used grain boundary (GB) to tune the HER activity of the catalyst. The engineered catalyst shows a very impressive HER activity and stability in alkaline and acidic electrolytes—outperforming the benchmark Pt/C catalyst. Some insights into the mechanism of the enhanced HER performance is provided. This work is very interesting and suitable for publication in Nat. Commun after the following comments are addressed.

The turn over frequency in KOH is relatively low. Can the authors explain the reason behind these lower values compared to those from the acidic electrolyte? I expect the TOF to be lower in basic electrolyte, but not like this.

The Tafel slope in alkaline electrolyte decreases with catalyst wt% and increases after. This is different from the behavior in acidic electrolyte. Is there a relationship between this trend and the mechanism of the HER?

The trends in onsets and overpotentials in both electrolytic solutions are different from those observed in the Tafel analyses. Have the authors further investigated the kinetics of the electrocatalyst? Are there kinetics' limitations with the electrocatalysts wt%?

The authors mentioned that Mo₂C derived from ternary compounds with termination groups OH show sluggish kinetics towards HER, but this is not necessarily true (<https://doi.org/10.1021/acscenergylett.6b00247>). Please revise this.

The variation of GB with the water content is interesting. Are there any correlations between this behavior and the electrocatalytic properties? Is there an optimum GB?

The presence of oxides does not seem to affect the stability of the material. Have the authors confirmed the oxidation states obtained from XPS with XANES? It would be informative to look at the catalyst before and after the reaction under XANES.

The lateral layer is relatively short compared to common nanosheets, especially those obtained from chemical exfoliation. Do the authors have a control over this? I suppose the active sites are located on the basal planes. Thus, large lateral size will lead to more catalytic activity.

The correlation between the Mo orbital energy level and H ads is interesting. However, more fundamental insights into the mechanism of HER will be useful. Have the authors explored this using

DFT calculations? These calculations must be done in aqueous environment to take into account the solvation effect.

Response to the reviewers

Reviewer #1 (Remarks to the Author):

In the current work, the authors said that their dense grain bounds from Mo₂C nanosheets afford an ultra-high density of active sites for HER. But they did not think about (1) the evolution of nitrogen concentration in N-doped graphene and (2) the effect of nitrogen and oxygen content in Mo₂C. All these can affect the hydrogen evolution reaction in this system. The work did not support the conclusions. There are many flaws in the data analysis, interpretation, and conclusions. This manuscript is prohibited from being published.

- We thank the reviewer for constructive comments, i.e., concentration aspect of nitrogen and oxygen. These questions have been largely discussed and corrected in our revised version, and all changes are highlighted in blue throughout the revision.

Yes, you are right. The evolution of nitrogen concentration in N-doped graphene and the nitrogen and oxygen content in Mo₂C can affect the HER activity.

- (1) **In this system, the evolution of the nitrogen concentration in graphene can be ignored to determine that the grain boundaries in Mo₂C nanosheets afford HER active sites.** As shown in **Supplementary Figs. 27, 28**, although the nitrogen concentration in graphene for Mo₂C/NG is higher than that of H-Mo₂C/NG, the types and proportions of nitrogen species in both hybrids are basically the same. Moreover, DFT calculations show that the free energies of hydrogen adsorption on graphitic-N and pyridinic-N are 0.74 and 0.85 eV (1), respectively, which are still much larger than that of Mo₂C (−0.405 eV) (2). These results indicate that the Mo₂C dominates the HER activity, and the contribution of N-doped graphene to HER can be ignored. Furthermore, H-Mo₂C/NG with a low nitrogen concentration in graphene possesses a soaring HER activity compared with Mo₂C/NG with a high nitrogen concentration, indicating that the dense grain boundaries in Mo₂C nanosheets afford an ultra-high density of active sites for HER. Besides, raw N-doped graphene (NG) is also prepared without (NH₄)₆Mo₇O₂₄·4H₂O precursor by the same procedure of Mo₂C/NG. Under acidic and alkaline conditions, the HER activity of NG is much lower than that of Mo₂C/NG and H-Mo₂C/NG (as shown in the figure below), which also proves that the evolution of nitrogen concentration in graphene has a negligible effect on the HER activity of the prepared Mo₂C hybrids in this work. We have added the discussion in the revised Supporting Information: “Although the nitrogen concentration in graphene for Mo₂C/NG is higher than that of H-Mo₂C/NG (**Supplementary Fig. 27**), the types and proportions of nitrogen species in both hybrids are basically the same. Meanwhile, DFT calculations show that the free energies of hydrogen adsorption on graphitic-N and pyridinic-N are 0.74 and 0.85 eV⁸, respectively, which are still much larger than that of Mo₂C (−0.405 eV)⁷. These results indicate that the Mo₂C nanocrystals dominate the HER activity, and the effect of the slight nitrogen concentration difference in graphene on the activity can be entirely ignored.” (Page 32)

Polarization curves of H-Mo₂C/NG, Mo₂C/NG, and NG in **a** 0.5 M H₂SO₄ and **b** 1.0 M KOH solutions.

- (2) Thank you for useful comments to improve the quality of this paper. Yes, the effect of nitrogen and oxygen content in Mo₂C on HER activity should be discussed. As shown in **Fig. 4b**, Mo 3d spectra of H-Mo₂C/NG and Mo₂C/NG are deconvoluted into four typical oxidation states (Mo²⁺, Mo³⁺, Mo⁴⁺, Mo⁶⁺) according to the probable existence of molybdenum species, in which the Mo²⁺ and Mo³⁺ can be ascribed to molybdenum carbides and nitrides, respectively, and the remaining Mo⁴⁺ and Mo⁶⁺ can be attributed to MoO₂ and MoO₃. MoO₂ and MoO₃ are unavoidable for Mo₂C-based materials due to their gradual oxidation at the surface upon exposure to air (3, 4). Previous reports show that surface MoO₃ in Mo₂C is in situ reduced to MoO₂ species during HER (5). As indicated by DFT calculations, the in situ reduced surface with terminal Mo=O moieties has free energy of hydrogen adsorption (ΔG_H) values of -0.236, -0.238, -0.260, and -0.266 eV on Mo sites closer to thermodynamic neutrality, which are better than those of bare Mo₂C (101) (-0.329 eV), MoO₂ (1.131 eV), Mo-O-Mo moieties (-0.421, -0.457, -0.542, and -0.395 eV), O=Mo=O, O=Mo-O-Mo, and Mo-O-Mo-O-Mo moieties (most of ΔG_H values for the models introducing two oxygen atoms are quite negative or positive). Although ΔG_H values for reduced Mo₂C surfaces with terminal Mo=O moieties are closer to thermodynamic neutrality among all possible Mo-O models, these values are not comparable to those at grain boundaries (such as -0.005, -0.010, -0.013, -0.044, 0.035, and 0.038 eV) for our H-Mo₂C/NG system (**Fig. 6e** and **Supplementary Figs. 53, 54**). Meanwhile, experiments show that MoO₂ and MoO₃ are not efficient catalysts for HER (6, 7). In detail, MoO₂ and MoO₃ require extremely large overpotentials of about 430 and 780 mV at pH = 0, and 440 and 660 mV at pH = 14 to reach a current density of 10 mA cm⁻², respectively. **Therefore, the relevant oxides in Mo₂C have some contribution to the electrocatalytic HER activity, but the effect of oxygen content in Mo₂C is negligible.** We have added the discussion in the revised manuscript: “Mo⁴⁺ and Mo⁶⁺ can be attributed to MoO₂ and MoO₃ caused by the oxidation of molybdenum species in ambient atmosphere, respectively, both of which are thought to be inactive for HER²⁰.” (Page 21)

Additionally, previous reports (8, 9) have revealed that the ΔG_H increases close to zero with increasing nitrogen content, which reflects the weakening of Mo-H bonding. However, with excessive N-doping, a positive ΔG_H (0.165 eV) can be observed, indicating that the surface bond H strength becomes too weak to capture H. And the doped nitrogen atom can't modify the HER activity while the adsorption site is far from nitrogen atom, suggesting the number of the active sites remained nearly unchanged. These results indicate that the enhancement of HER activity at low nitrogen content is not significant. In this work, the atomic percentage of Mo³⁺ in H-Mo₂C/NG is

10.01%, which is slightly higher than that of Mo₂C/NG (8.83%) (**Fig. 4b**). However, compared with Mo₂C/NG, H-Mo₂C/NG affords sharply enhanced HER activities in both acidic and alkaline electrolytes (**Fig. 5**). Meanwhile, previous reports have confirmed that slight differences of nitrogen content in Mo₂C have little effect on the electrocatalytic HER activity (8-11). For example, when the nitrogen content in Mo₂C increases from 6.37% to 10.83%, the overpotential merely decreases from 535 to 529 mV in 0.5 M H₂SO₄ and from 517 to 482 mV in 1.0 M KOH at a current density of 100 mA cm⁻² (2). **Therefore, in this work, the effect of nitrogen content in Mo₂C on HER activity for Mo₂C hybrids can be ignored, and dense grain boundaries in Mo₂C nanosheets afford an ultra-high density of active sites for HER.** We have added the discussion in the revised manuscript and revised Supporting Information: “In N 1s spectra, the intensity of Mo–N species increases slightly with the increase of water content (**Supplementary Fig. 30**), indicating that a certain water can also promote N-doping.” (Page 22 in revised manuscript) “Although the introduction of nitrogen atoms into Mo₂C is beneficial for electrocatalytic HER, the nitrogen species in Mo₂C have very little effect on HER activity⁵¹⁻⁵³. Compared with Mo₂C/NG, H-Mo₂C/NG with abundant GBs affords abruptly enhanced HER activity, which indicates that the activity mainly originates from the GBs in Mo₂C NSs.” (Page 28 in revised manuscript) “The atomic percentages of Mo³⁺ in H-Mo₂C/NG and Mo₂C/NG are 10.01% and 8.83% (**Fig. 4b**), respectively.” (Page 33 in revised Supporting Information)

We are sorry that there are lots of questions that we are not explained clear in the previous version. All the insightful suggestions the reviewer raised have been thoroughly considered, and the corresponding revisions have been made. Again, we truly appreciate the reviewer’ careful review and kind guidance on this work, which is vital for us to improve quality of this manuscript. We wish the revision fulfills your requirements to be published in *Nature Communications*.

References

1. Wang, Z. et al. Theoretical calculation guided electrocatalysts design: Nitrogen saturated porous Mo₂C nanostructures for hydrogen production. *Appl. Catal. B* **257**, 117891 (2019).
2. Yan, H. et al. Holey reduced graphene oxide coupled with an Mo₂N–Mo₂C heterojunction for efficient hydrogen evolution. *Adv. Mater.* **30**, 1704156 (2018).
3. Wan, C., Regmi Y. N. & Leonard B. M. Multiple phases of molybdenum carbide as electrocatalysts for the hydrogen evolution reaction. *Angew. Chem. Int. Ed.* **53**, 6407–6410 (2014).
4. Ma, R. et al. Ultrafine molybdenum carbide nanoparticles composited with carbon as a highly active hydrogen-evolution electrocatalyst. *Angew. Chem. Int. Ed.* **54**, 14723–14727 (2015).
5. He, L. et al. Molybdenum carbide-oxide heterostructures: In situ surface reconfiguration toward efficient electrocatalytic hydrogen evolution. *Angew. Chem. Int. Ed.* **59**, 3544–3548 (2020).
6. Li, J.-S. et al. Coupled molybdenum carbide and reduced graphene oxide electrocatalysts for efficient hydrogen evolution. *Nat. Commun.* **7**, 11204 (2016).
7. Vrubel, H. & Hu, X. Molybdenum boride and carbide catalyze hydrogen evolution in both acidic and basic solutions. *Angew. Chem. Int. Ed.* **51**, 12703–12706 (2012).
8. Huang, Y. et al. Fine tuning electronic structure of catalysts through atomic engineering for enhanced hydrogen evolution. *Adv. Energy Mater.* **8**, 1800789 (2018).
9. Wang, Z. et al. Theoretical calculation guided electrocatalysts design: Nitrogen saturated porous Mo₂C nanostructures for hydrogen production. *Appl. Catal. B* **257**, 117891 (2019).
10. Li, S. et al. Metal-organic precursor-derived mesoporous carbon spheres with homogeneously distributed

molybdenum carbide/nitride nanoparticles for efficient hydrogen evolution in alkaline media. *Adv. Funct. Mater.* **29**, 1807419 (2019).

11. Huang, Y. et al. Nitrogen-doped porous molybdenum carbide and phosphide hybrids on a carbon matrix as highly effective electrocatalysts for the hydrogen evolution reaction. *Adv. Energy Mater.* **8**, 1701601 (2018).

1. References are not in the correct positions. Such as reference 9.

- We thank the reviewer for the suggestion. We have carefully checked and corrected references in revised manuscript.

2. Should prove the active sites for HER are in the grain boundary from experiments and calculations if their conclusion is correct.

- We really appreciate your comments. **Experimentally**, TEM and IPF images intuitively show that the grain boundary density is very sensitive to water. With the increase of water content, the structure of Mo₂C evolves from nanoparticles (Mo₂C/NG) (**Supplementary Fig. 2**) to discontinuous nanosheets (H-Mo₂C/NG-7.14) (**Supplementary Fig. 16**), then to continuous nanosheets with rich grain boundaries (H-Mo₂C/NG) (**Supplementary Fig. 17**), and finally to layered stacked sheets (H-Mo₂C/NG-43.28) (**Supplementary Fig. 18**). The grain boundary density of Mo₂C nanosheets in Mo₂C hybrids is summarized in **Supplementary Fig. 8**, where the statistical region of each sample is ~ 2,0000 nm² and H-Mo₂C/NG possesses the highest grain boundary density of $138 \pm 3 \mu\text{m}^{-1}$. Meanwhile, Mo 3d spectra of Mo₂C hybrids are deconvoluted into four typical oxidation states (Mo²⁺, Mo³⁺, Mo⁴⁺, Mo⁶⁺) (**Fig. 4b** and **Supplementary Fig. 29**), in which the Mo²⁺ and Mo³⁺ can be ascribed to molybdenum carbides and nitrides, respectively, and the remaining Mo⁴⁺ and Mo⁶⁺ can be attributed to MoO₂ and MoO₃. Despite the presence of nitrides and oxides, these low-activity molybdenum byproducts and their minor content differences have a negligible effect on the HER activity of the Mo₂C hybrids (1-3). Particularly, as the water content increases, the η_{10} and Tafel slopes on Mo₂C hybrids firstly decrease and then increase in acidic (**Supplementary Figs. 34, 35**) and alkaline (**Supplementary Figs. 36, 37**) electrolytes, where H-Mo₂C/NG achieves the best HER activity with water content of 18.56 wt%. **Thus, grain boundaries vary with water content, and the grain boundary has a certain correlation with the electrocatalytic HER performance; that is, the higher the grain boundary density, the better the electrocatalytic performance.** The above results reveal that the active sites for HER are located at the grain boundaries of Mo₂C nanosheets. Furthermore, **DFT calculations** also show optimized H adsorption free energies (such as -0.005, -0.010, -0.013, -0.044, 0.035, and 0.038 eV) at the fcc/hcp, fcc/fcc, and hcp/hcp grain boundaries (**Fig. 6e** and **Supplementary Figs. 53, 54**), which are far superior to those of sites far from the grain boundaries (such as -0.324, -0.358, -0.428, and -0.517 eV). Therefore, the abundant grain boundaries in Mo₂C nanosheets provide the high active sites for HER and jointly promote the electrocatalytic hydrogen production.

References

1. He, L. et al. Molybdenum carbide-oxide heterostructures: In situ surface reconfiguration toward efficient electrocatalytic hydrogen evolution. *Angew. Chem. Int. Ed.* **59**, 3544–3548 (2020).
2. Yan, H. et al. Holey reduced graphene oxide coupled with an Mo₂N–Mo₂C heterojunction for efficient

hydrogen evolution. *Adv. Mater.* **30**, 1704156 (2018).

- Wang, Z. et al. Theoretical calculation guided electrocatalysts design: Nitrogen saturated porous Mo₂C nanostructures for hydrogen production. *Appl. Catal. B* **257**, 117891 (2019).

Reviewer #2 (Remarks to the Author):

We are grateful for the reviewer's professional comments and suggestions. We are sorry that there are lots of questions that we are not explained clear in the previous version. We have improved the manuscript (all changes highlighted in blue throughout the revision) and the replies to your comments are given below. We wish the revision fulfills your requirements to be published in *Nature Communications*.

1. What are the noteworthy results?

Good HER performance, simple synthesis methods and reagents. They claim grain boundary control however these results are questionable.

- We sincerely appreciate the reviewer's insightful and professional comments. Those comments are all very valuable and helpful for us to revise and improve this manuscript. In this work, combining the results of AFM (**Fig. 2a, c**) and TEM (**Fig. 2b, d**) images confirm beyond doubt that water can induce the morphological evolution of Mo₂C from nanoparticles to nanosheets during carburization process. Meanwhile, DFT calculations also reveal the above-mentioned growth behavior of Mo₂C nanocrystals with the assistance of water. That is, common MoO₃ without H₂O can only react with carbon on reduced graphene oxide edge, producing Mo₂C nanoparticles. While moist MoO₃ with H₂O not only reacts with the edge C atoms, but also readily combines with six-membered C-ring along the reduced graphene oxide basal plane; in this situation, epitaxial growth is carried out to enlarge the Mo₂C domains, achieving a coalesced few-layer nanosheets with several micrometers in lateral size. Furthermore, TEM and IPF images intuitively show that the grain boundary density is very sensitive to water. With the increase of water content, the structure of Mo₂C evolves from nanoparticles (Mo₂C/NG) (**Supplementary Fig. 2**) to discontinuous nanosheets (H-Mo₂C/NG-7.14) (**Supplementary Fig. 16**), then to continuous nanosheets with rich grain boundaries (H-Mo₂C/NG) (**Supplementary Fig. 17**), and finally to layered stacked sheets (H-Mo₂C/NG-43.28) (**Supplementary Fig. 18**). Meanwhile, the grain boundary density of Mo₂C nanosheets in Mo₂C hybrids is summarized in **Supplementary Fig. 8**, where the statistical region of each sample is ~ 2,0000 nm². Obviously, H-Mo₂C/NG possesses the highest grain boundary density of $138 \pm 3 \mu\text{m}^{-1}$, clearly demonstrating that the water concentration in MoO₃/RGO intermediates effectively manipulates the grain boundary density of Mo₂C nanosheets.

Additionally, sufficient oxygen-containing functional groups on the graphene oxide surface are necessary for the formation of high-density grain boundaries in Mo₂C nanosheets. Specifically, due to the limited nucleation sites of graphene oxide with low oxidation degree, Mo₂C nanosheets with a small amount grain boundaries are obtained (**Supplementary Fig. 11**). While the morphology of Mo₂C product supported with excessively oxidized graphene oxide is slightly agglomerated (**Supplementary Fig. 12**) as compared with the uniformly structure of H-Mo₂C/NG synthesized with common graphene oxide as substrate (**Fig. 2d**). During the carburization process, the escape of a large number of oxygen-containing functional groups on graphene severely damages the framework structure, thereby reducing the grain boundary density in Mo₂C nanosheets. Therefore, the grain boundary density in Mo₂C nanosheets can be regulated by controlling the water content of the MoO₃/RGO intermediate and the oxidation degree

of graphene oxide. When common graphene oxide is used as the substrate and the water content of the intermediate is 18.56 wt%, the structurally optimized Mo₂C nanosheets with the highest grain boundary density of $138 \pm 3 \mu\text{m}^{-1}$ are obtained.

2. Will the work be of significance to the field and related fields? How does it compare to the established literature? If the work is not original, please provide relevant references.

Many groups have investigated the synthesis of molybdenum nitride and carbide through reaction of molybdenum oxide and ammonia or ammonolysis. In addition, there is a vast field of Mo₂C catalysts using support materials like graphene and rGO. This paper is unique in observing the effects that water content have on the overall reaction and their investigation provide some new insights or understudied to the field. However they also need to include the role that NH₃ plays in the reaction, and if its affects play a role in the formation of nanoparticles, nanosheets, and/or grain boundaries. I feel like the results and discussion in their current form do not justify this as a significant advancement of the field.

➤ We appreciate your suggestions for improving the quality of this article. Yes, many groups have investigated the synthesis of molybdenum nitride and carbide through reaction of molybdenum oxide and ammonia or ammonolysis. In addition, there is a vast field of Mo₂C catalysts using support materials like graphene and rGO. **However, a simple and general strategy to manipulate grain boundary density of Mo₂C nanosheets in powdered nanomaterials and explore the intrinsic relationship between grain boundaries and HER activity have not been reported.** In this manuscript, we demonstrate a general water-assisted carbothermal reaction strategy without any gaseous carbon feeding for the construction of ultrathin Mo₂C nanosheets with high-density grain boundaries supported on N-doped graphene. During carbonization process, water induces the structural evolution of Mo₂C nanocrystals from nanoparticles to nanosheets and controls the grain boundary density of Mo₂C nanosheets as well. The dense polycrystalline Mo₂C nanosheets with massive grain boundaries are covalently connected with underneath N-doped graphene substrates through Mo–C bonds, which affords an ultra-high density of active sites for HER, giving Pt-like HER activity and superior electrocatalytic stability. DFT calculations reveal that the d_z^2 orbital energy level of metal atoms exhibits an intrinsic relationship with the catalyst activity and is regarded as a new descriptor for accurately predicting the HER activity.

The concern about the role of NH₃ is reasonable, and the role of NH₃ in the reaction to generate Mo₂C nanosheets with rich grain boundary should be discussed. We have added the discussion in the revised manuscript: “Moreover, the Mo–Mo₂C heterostructures are formed by the same procedure of H–Mo₂C/NG without NH₃ (**Supplementary Fig. 32**), indicating that NH₃ is essential for harvesting pure Mo₂C crystalline phase. The incomplete carbonization of MoO₃ intermediate in the absence of NH₃ possibly results from the sluggish carburation kinetics induced by the slow pyrolysis process of RGO³⁸. When NH₃ is introduced into the carbonization process, the nitrogen atoms from the pyrolysis of NH₃ can be quickly supplied to the reduced molybdenum atoms³⁹, preventing the formation of metallic molybdenum, resulting in trace nitrogen-doped Mo₂C NSs with dense GBs.” (Page 22) We wish the revision fulfills your requirements to be published in *Nature Communications*.

The concern about the role of NH₃ is reasonable, and the role of NH₃ in the reaction to generate Mo₂C nanosheets with rich grain boundary should be discussed. We have added the discussion in the revised manuscript: “Moreover, the Mo–Mo₂C heterostructures are formed by the same procedure of H–Mo₂C/NG without NH₃ (**Supplementary Fig. 32**), indicating that NH₃ is essential for harvesting pure Mo₂C crystalline phase. The incomplete carbonization of MoO₃ intermediate in the absence of NH₃

possibly results from the sluggish carburation kinetics induced by the slow pyrolysis process of RGO³⁸. When NH₃ is introduced into the carbonization process, the nitrogen atoms from the pyrolysis of NH₃ can be quickly supplied to the reduced molybdenum atoms³⁹, preventing the formation of metallic molybdenum, resulting in trace nitrogen-doped Mo₂C NSs with dense GBs.” (Page 22) We wish the revision fulfills your requirements to be published in *Nature Communications*.

Regarding the current results and discussions that do not justify this as a significant advancement of the field, we slightly disagree to the reviewer’s point. Combining experiments (Fig. 2a-d) and DFT calculations (Fig. 1) reveal that water, with the assistance of NH₃, can induce the morphology transition of Mo₂C from nanoparticles to nanosheets during carbonization process. Controlling the content of introduced water can manipulate the density of grain boundaries in the Mo₂C nanosheets (Supplementary Figs. 8, 16-18), thereby regulating the electrocatalytic HER activity (Fig. 5 and Supplementary Figs. 34-37). The general synthetic strategy of nanosheets with rich grain boundaries can also be extended to other transition metal carbide systems (Supplementary Fig. 47), such as vanadium (Supplementary Fig. 48), niobium (Supplementary Fig. 49), tantalum (Supplementary Fig. 50), and tungsten (Supplementary Fig. 51), and yield excellent HER performances in pH-universal electrolyte (Supplementary Fig. 52), demonstrating the generality of the water-assisted carbothermal reaction strategy. This work opens up a new avenue for accelerating development of high-efficiency catalysts through grain boundary engineering. Furthermore, DFT calculations offer important insight into the correlation between grain boundaries and the enhanced reactivity. The d_z² orbital energy level of metal atoms exhibits an intrinsic relationship with the catalyst activity and is regarded as a new descriptor for accurately predicting the HER activity (Fig. 6e). Compared with the state-of-the-art descriptors such as band-energy theory and work function that describe the properties of the whole surface, the d_z² orbital energy level is more relevant to the active site of the catalysts. Our finding broadens the applicability of d band theory to activity prediction of metal electrocatalysts. **Taken together, we believe that the current results and discussions can demonstrate that the generality of the water-assisted carbothermal reaction strategy and the new descriptor of HER activity are significant advancements in the field.**

More details are provided in revised Supplementary Information (Supplementary Fig. 32). (Page 36)

Supplementary Fig. 32. XRD pattern of Mo-Mo₂C heterostructures on RGO. The XRD pattern reveal the formation of Mo₂C nanocrystals and metallic Mo compositions by the same procedure as H-Mo₂C/NG in the absence of NH₃.

3. Does the work support the conclusions and claims, or is additional evidence needed?

In short, no. There are some nice results and good data overall, however, I think determining vital information about the grain boundaries and the mechanism of their formation are quite difficult and not necessarily supported by the data. In addition, they should include much more discussion on the amount of Mo-N or molybdenum nitride present and the role that NH₃ plays in these reactions. Heating MoO₃ in NH₃ likely forms nitride which converts to the carbide in the presence of carbon (rGO). They claim the NH₃ is needed to reduce the rGO and form nitrogen doping, however, they barely mention that there is Mo-N other than the XPS section. If they anneal in absence of NH₃ or use ammonia in the hydrothermal portion of the synthesis instead of the high temperature furnace they can avoid the formation of the nitride and validate that it is simply rGO and Mo₂C not Mo-N responsible for their catalytic results.

➤ We appreciate the reviewer's professional comments. Yes, determining vital information about grain boundaries and the mechanism of their formation is quite difficult. In this work, we combine experimental observations and theoretical calculations to uncover the formation mechanism of grain boundaries in Mo₂C nanosheets. **Experimentally**, we observe that the grain boundary density in Mo₂C nanosheets is strongly dependent on the water content of the MoO₃ intermediate and the oxidation degree of graphene oxide. As shown in the TEM and IPF images, with the increase of water content, the structure of Mo₂C evolves from nanoparticles (Mo₂C/NG) (**Supplementary Fig. 2**) to discontinuous nanosheets (H-Mo₂C/NG-7.14) (**Supplementary Fig. 16**), then to continuous nanosheets with rich grain boundaries (H-Mo₂C/NG) (**Supplementary Fig. 17**), and finally to layered stacked sheets (H-Mo₂C/NG-43.28) (**Supplementary Fig. 18**). The grain boundary density of Mo₂C nanosheets in Mo₂C hybrids is summarized in **Supplementary Fig. 8**. Obviously, H-Mo₂C/NG possesses the highest grain boundary density of $138 \pm 3 \mu\text{m}^{-1}$, distinctly proving that the water content in MoO₃/RGO effectively manipulates the grain boundary density of Mo₂C nanosheets. Meanwhile, sufficient oxygen-containing functional groups on the graphene oxide surface are necessary for the formation of high-density grain boundaries in Mo₂C nanosheets. Specifically, Mo₂C nanosheets with few grain boundaries are obtained due to the limited nucleation sites of graphene oxide with low oxidation degree (**Supplementary Fig. 11**). While the morphology of Mo₂C supported with excessively oxidized graphene oxide is slightly agglomerated (**Supplementary Fig. 12**) as compared with the uniformly structure of H-Mo₂C/NG synthesized with common graphene oxide as substrate (**Fig. 2d**). During the carburization process, the escape of a large number of oxygen-containing functional groups on graphene oxide severely damages the framework structure, thereby reducing the grain boundary density in Mo₂C nanosheets. Furthermore, **theoretical calculations** reveal that the common MoO₃ without H₂O can only react with carbon on RGO edge, producing Mo₂C nanoparticles. While moist MoO₃ with H₂O not only reacts with the edge C atoms, but also readily combines with six-membered C-ring along the RGO basal plane; in this situation, epitaxial growth is carried out on to enlarge the Mo₂C domains, achieving a coalesced nanosheets with rich grain boundaries. Therefore, in the presence of water, due to the guest–host interaction between H₂O and MoO₃, unique Mo–O configurations are obtained and anchored on RGO through strong charge transfer, followed by carbothermal reduction, which serves as the nucleation site for Mo₂C growth. As the activated carbon species encounter Mo–O configurations on RGO sheets, the dense seeds begin to individually nucleate at multiple regions. Consequently, these simultaneously grown nanocrystals merge together seamlessly during the chemical coalescence process and eventually form Mo₂C nanosheets, which thereby hinders the growth of Mo₂C grains in the radial orientation, leaving behind numerous grain boundaries.

We agree with the reviewer's comments that the amount of Mo–N and the role of NH₃ in the reaction should be discussed. We have added the discussion in the revised manuscript and revised Supporting Information: “In N 1s spectra, the intensity of Mo–N species increases slightly with the increase of water content (**Supplementary Fig. 30**), indicating that a certain water can also promote N-doping.” (Page 22 in revised manuscript) “Moreover, the Mo–Mo₂C heterostructures are formed by the same procedure of H–Mo₂C/NG without NH₃ (**Supplementary Fig. 32**), indicating that NH₃ is essential for harvesting pure Mo₂C crystalline phase. The incomplete carbonization of MoO₃ intermediate in the absence of NH₃ possibly results from the sluggish carburation kinetics induced by the slow pyrolysis process of RGO³⁸. When NH₃ is introduced into the carbonization process, the nitrogen atoms from the pyrolysis of NH₃ can be quickly supplied to the reduced molybdenum atoms³⁹, preventing the formation of metallic molybdenum, resulting in trace nitrogen-doped Mo₂C NSs with dense GBs.” (Page 22 in revised manuscript) “Although the introduction of nitrogen atoms into Mo₂C is beneficial for electrocatalytic HER, the nitrogen species in Mo₂C have very little effect on HER activity⁵¹⁻⁵³. Compared with Mo₂C/NG, H–Mo₂C/NG with abundant GBs affords abruptly enhanced HER activity, which indicates that the activity originates from the GBs in Mo₂C NSs.” (Page 28 in revised manuscript) “The atomic percentages of Mo³⁺ in H–Mo₂C/NG and Mo₂C/NG are 10.01% and 8.83% (**Fig. 4b**), respectively.” (Page 33 in revised Supporting Information)

In addition, according to the existing conclusions reported by Wang et al. (1-3), if the hydrothermal portion of the synthesis uses nitrogen sources instead of a high temperature furnace, the intermediate product is not pure MoO₃. **In this work, DFT calculations prove that the interaction of MoO₃ intermediate with water to obtain a unique Mo–O configuration is a necessary condition for the formation of Mo₂C nanosheets.** Thus, introducing nitrogen sources during the hydrothermal reaction is unable to obtain grain boundary-rich Mo₂C nanosheets without nitrogen atom doping. Although the introduction of nitrogen atoms into Mo₂C is beneficial for electrocatalytic HER, the nitrogen species in Mo₂C have very little effect on HER activity (1, 2, 4). For example, when the nitrogen content in Mo₂C increases from 6.37% to 10.83%, the overpotential merely decreases from 535 to 529 mV in 0.5 M H₂SO₄ and from 517 to 482 mV in 1.0 M KOH at a current density of 100 mA cm⁻² (4). Therefore, the introduction of nitrogen species in Mo₂C nanosheets does not prevent the identification of HER active sites originating from dense grain boundaries.

References

1. Wang, Z. et al. Theoretical calculation guided electrocatalysts design: Nitrogen saturated porous Mo₂C nanostructures for hydrogen production. *Appl. Catal. B* **257**, 117891 (2019).
2. Li, S. et al. Metal-organic precursor-derived mesoporous carbon spheres with homogeneously distributed molybdenum carbide/nitride nanoparticles for efficient hydrogen evolution in alkaline media. *Adv. Funct. Mater.* **29**, 1807419 (2019).
3. Anjum, M. A. R., Lee, M. H. & Lee, J. S. Boron-and nitrogen-codoped molybdenum carbide nanoparticles imbedded in a BCN network as a bifunctional electrocatalyst for hydrogen and oxygen evolution reactions. *ACS Catal.* **8**, 8296–8305 (2018).
4. Yan, H. et al. Holey reduced graphene oxide coupled with an Mo₂N–Mo₂C heterojunction for efficient hydrogen evolution. *Adv. Mater.* **30**, 1704156 (2018).

4. Are there any flaws in the data analysis, interpretation and conclusions? Do these prohibit publication or require revision?

Yes

(1) I don't agree that there is a CVD process going on. It seems like the precursor solid has the carbon and Mo present and then they perform an ammonolysis reaction on the MoO₃ which has been used for decades to make nitrides.

- We appreciate your comments for improving the quality of this article. Yes, the process from MoO₃ intermediate to Mo₂C product should be more accurately described as a carbonization process where a carbothermic reduction reaction occurs (1). Carbothermal reactions are thermal reactions that use carbon as the reducing agent at high temperature (2-4), in which most metal oxide can be reduced into metal and metal carbide accompanying with the carbon atom leaving in the form of carbon monoxide and carbon dioxide (5). The most prominent example is iron ore smelting, which is used in the production of iron or steel. In our case, although the process of introducing the necessary NH₃ and forming a small amount of nitrogen-doped product during the carbonization of MoO₃ can be assigned to the CVD process (6, 7), the process of constructing the dominant Mo₂C product is a carbonization process. Therefore, in this work, the carbonization process is based on the direct reaction between MoO₃ intermediate and reduced graphene oxide substrate without any carbon gas feeding, enabling Mo₂C hybrids have greatly strengthened interaction between Mo₂C nanosheets and N-doped graphene sheets through Mo–C bonds, which are high reproducibility in terms of structure and performance. **We have corrected all inaccurate expressions about the CVD process in the revised manuscript.** We wish the revision fulfills your requirements to be published in *Nature Communications*.

References

1. Zhou, D. et al. A general and scalable synthesis approach to porous graphene. *Nat. Commun.* **5**, 4716 (2014).
2. Yoshikawa, T. & Morita, K. Carbothermic reduction of MgO by microwave irradiation. *Mater. Trans.* **44**, 722–726 (2003).
3. Setoudeh, N., Saidi, A. & Welham, N. J. Carbothermic reduction of anatase and rutile. *J. Alloys Comp.* **390**, 138–143 (2005).
4. Li, S. et al. Enrichment of semiconducting single-walled carbon nanotubes by carbothermic reaction for use in all-nanotube field effect transistors. *ACS Nano* **6**, 9657–9661 (2012).
5. Li, X., Ma, D., Chen, L. & Bao, X. Fabrication of molybdenum carbide catalysts over multi-walled carbon nanotubes by carbothermal hydrogen reduction. *Catal. Lett.* **116**, 63–69 (2007).
6. Jia, J. et al. Ultrathin N-doped Mo₂C nanosheets with exposed active sites as efficient electrocatalyst for hydrogen evolution reactions. *ACS Nano* **11**, 12509–12518 (2017).
7. Wang, H. et al. Elastic properties of 2D ultrathin tungsten nitride crystals grown by chemical vapor deposition. *Adv. Funct. Mater.* **29**, 1902663 (2019).

(2) They rarely mention the fact that Mo-nitride is a significant part of the product and do not account for it in the explanation of the results. Nitride is only mentioned once in the document.

- We appreciate the reviewer's important comments. Yes, the Mo–N component in the products should be discussed more, and we have added the discussion in the revised manuscript and revised Supporting Information: "In N 1s spectra, the intensity of Mo–N species increases slightly with the increase of water content (**Supplementary Fig. 30**), indicating that a certain water can also promote N-doping." (Page 22

in revised manuscript) “Although the introduction of nitrogen atoms into Mo₂C is beneficial for electrocatalytic HER, the nitrogen species in Mo₂C have very little effect on HER activity⁵¹⁻⁵³. Compared with Mo₂C/NG, H-Mo₂C/NG with abundant GBs affords abruptly enhanced HER activity, which indicates that the activity mainly originates from the GBs in Mo₂C NSs.” (Page 28 in revised manuscript) “The atomic percentages of Mo³⁺ in H-Mo₂C/NG and Mo₂C/NG are 10.01% and 8.83% (**Fig. 4b**), respectively.” (Page 33 in revised Supporting Information)

(3) The role of ammonia should also be described in more detail. Do they get the same number of grain boundaries using inert atmosphere? Does going through a nitride intermediate make a difference on the carbide formation, particle size, grain boundaries etc

➤ We appreciate your suggestions for improving the quality of this article. You are right, the role of ammonia should also be described in more detail. We have added the described in the revised manuscript: “Moreover, the Mo-Mo₂C heterostructures are formed by the same procedure of H-Mo₂C/NG without NH₃ (**Supplementary Fig. 32**), indicating that NH₃ is essential for harvesting pure Mo₂C crystalline phase. The incomplete carbonization of MoO₃ intermediate in the absence of NH₃ possibly results from the sluggish carburation kinetics induced by the slow pyrolysis process of RGO³⁸. When NH₃ is introduced into the carbonization process, the nitrogen atoms from the pyrolysis of NH₃ can be quickly supplied to the reduced molybdenum atoms³⁹, preventing the formation of metallic molybdenum, resulting in trace nitrogen-doped Mo₂C NSs with dense GBs.” (Page 22) Additionally, in an inert atmosphere, Mo-Mo₂C heterostructures are obtained instead of pure Mo₂C crystalline phase, so in this case, discussing the number of grain boundaries formed by stitching between Mo₂C grains in the Mo-Mo₂C heterostructures is not reliable.

(4) They also need to include the nitride and any relevant oxides in the DFT calculations to further explain if they play a role in the catalytic activity and stability of these materials.

➤ We thank the reviewer for the professional suggestions. The role of nitride in the catalytic HER activity of Mo₂C-based materials has been widely discussed. DFT calculations (*I-2*) have revealed that the negative free energy of hydrogen adsorption (ΔG_H) increases close to zero with increasing nitrogen content, which reflects the weakening of Mo-H bonding. However, with excessive N-doping, a positive ΔG_H (0.165 eV) can be observed, indicating that the surface bond H strength becomes too weak to capture H. And the doped nitrogen atom can't modify the HER activity while the adsorption site is far from nitrogen atom, suggesting the number of the active sites remained nearly unchanged. These results indicate that the enhancement of HER activity at low nitrogen content is not remarkable. In this work, the atomic percentage of Mo³⁺ in H-Mo₂C/NG is 10.01%, which is slightly higher than that of Mo₂C/NG (8.83%) (**Fig. 4b**). However, compared with Mo₂C/NG, H-Mo₂C/NG affords sharply enhanced HER activities in both acidic and alkaline electrolytes (**Fig. 5**). Meanwhile, previous reports have confirmed that slight differences of nitrogen content in Mo₂C have little effect on the electrocatalytic HER activity (*I-5*). For example, when the nitrogen content in Mo₂C increases from 6.37% to 10.83%, the overpotential merely decreases from 535 to 529 mV in 0.5 M H₂SO₄ and from 517 to 482 mV in 1.0 M KOH at a current density of 100 mA cm⁻² (*5*). Furthermore, DFT calculations also reveal the optimized H adsorption free energies (such as -0.005, -0.010, -0.013, -0.044, 0.035, and 0.038 eV) at the fcc/hcp, fcc/fcc, and hcp/hcp grain boundaries for our H-Mo₂C/NG system (**Fig. 6e** and **Supplementary Figs.**

53, 54).

Therefore, in this work, the nitride in Mo₂C contributes to the electrocatalytic HER activity, but the dominant active sites come from the dense grain boundaries in Mo₂C nanosheets.

MoO₂ and MoO₃ are unavoidable for Mo₂C-based materials due to their gradual oxidation at the surface upon exposure to air. Previous reports show that surface MoO₃ in Mo₂C is in situ reduced to MoO₂ species during HER (6). Moreover, as indicated by DFT calculations, the in situ reduced surface with terminal Mo=O moieties has ΔG_H values of -0.236, -0.238, -0.260, and -0.266 eV on Mo sites closer to thermodynamic neutrality, which are better than those of bare Mo₂C (101) (-0.329 eV), Mo-O-Mo moieties (-0.421, -0.457, -0.542, and -0.395 eV), O=Mo=O, O=Mo-O-Mo, and Mo-O-Mo-O-Mo moieties (most of ΔG_H values for the models introducing two oxygen atoms are quite negative or positive). Although ΔG_H values for reduced Mo₂C surfaces with terminal Mo=O moieties are closer to thermodynamic neutrality among all possible Mo-O models, these values are not comparable to those at grain boundaries (such as -0.005, -0.010, -0.013, -0.044, 0.035, and 0.038 eV) for our H-Mo₂C/NG system (**Fig. 6e** and **Supplementary Figs. 53, 54**). **Thence, the relevant oxides in Mo₂C hybrids contribute little to the electrocatalytic activity (7), and the HER activity at the grain boundaries dominates.**

Additionally, due to the excellent corrosion resistance of molybdenum nitride (8-10), the nitride in Mo₂C does not affect the stability of the prepared Mo₂C hybrids for electrocatalytic hydrogen production. Meanwhile, molybdenum oxides often act as HER electrocatalysts in alkaline electrolytes due to their instability in acidic solutions (11). Moreover, the dominant activity in this work originates from Mo₂C nanosheets with rich grain boundaries that are extremely difficult to reduce during HER process (6). Thus, the nitride and inactive oxides in Mo₂C cannot affect the stability of the Mo₂C hybrids for electrocatalytic hydrogen production in acidic and alkaline solutions. As revealed in **Supplementary Fig. 43**, after 10,000 or even 50,000 cycles in 0.5 M H₂SO₄ and 1.0 M KOH, the shifts of polarization curves for H-Mo₂C/NG are negligible compared with the initial curves. Meanwhile, after 100,000 s of operation at $j = -100, -200, \text{ and } -300 \text{ mA cm}^{-2}$, the potentials of H-Mo₂C/NG only increase by 8, 14, and 38 mV in 0.5 M H₂SO₄, and 11, 13, and 36 mV in 1.0 M KOH, respectively (**Fig. 5g**). The XRD characterization displays that Mo₂C is well crystallized, and no new phase is experimentally detected (**Supplementary Fig. 45**). Furthermore, the Mo 3d spectrum of H-Mo₂C/NG after cycling test is unambiguously deconvoluted into four pairs peaks of Mo²⁺, Mo³⁺, Mo⁴⁺, and Mo⁶⁺ species, in which Mo₂C as active species still dominates (**Supplementary Fig. 46**). Besides, TEM images show that Mo₂C nanocrystals in H-Mo₂C/NG still maintain an ultrathin nanosheet structure with highly exposed high-density grain boundaries after electrochemical cycling (**Supplementary Fig. 44**). **As a result, combined with HER performance test, XRD, XPS, and TEM results after cycling, the H-Mo₂C/NG catalyst possesses excellent stability during the HER process, and the nitride and relevant oxides do not affect the stability of prepared Mo₂C hybrids.** The above results can already explain the role of nitride and relevant oxides in the catalytic activity and stability of Mo₂C hybrids, so we consider it is unnecessary to introduce these nitride and oxides in DFT calculations.

References

1. Huang, Y. et al. Fine tuning electronic structure of catalysts through atomic engineering for enhanced hydrogen evolution. *Adv. Energy Mater.* **8**, 1800789 (2018).
2. Wang, Z. et al. Theoretical calculation guided electrocatalysts design: Nitrogen saturated porous Mo₂C nanostructures for hydrogen production. *Appl. Catal. B* **257**, 117891 (2019).

3. Li, S. et al. Metal-organic precursor-derived mesoporous carbon spheres with homogeneously distributed molybdenum carbide/nitride nanoparticles for efficient hydrogen evolution in alkaline media. *Adv. Funct. Mater.* **29**, 1807419 (2019).
4. Huang, Y. et al. Nitrogen-doped porous molybdenum carbide and phosphide hybrids on a carbon matrix as highly effective electrocatalysts for the hydrogen evolution reaction. *Adv. Energy Mater.* **8**, 1701601 (2018).
5. Yan, H. et al. Holey reduced graphene oxide coupled with an Mo₂N–Mo₂C heterojunction for efficient hydrogen evolution. *Adv. Mater.* **30**, 1704156 (2018).
6. He, L. et al. Molybdenum carbide–oxide heterostructures: In situ surface reconfiguration toward efficient electrocatalytic hydrogen evolution. *Angew. Chem. Int. Ed.* **59**, 3544–3548 (2020).
7. Gao, Q., Zhang, W., Shi, Z., Yang, L. & Tang, Y. Structural design and electronic modulation of transition-metal-carbide electrocatalysts toward efficient hydrogen evolution. *Adv. Mater.* **31**, 1802880 (2019).
8. Zhu, J. et al. Recent advances in electrocatalytic hydrogen evolution using nanoparticles. *Chem. Rev.* **120**, 851–918 (2019).
9. Peng, X. et al. Recent progress of transition metal nitrides for efficient electrocatalytic water splitting. *Sustain. Energy Fuels* **3**, 366–381 (2019).
10. Guan, H. et al. Low temperature synthesis of plasmonic molybdenum nitride nanosheets for surface enhanced Raman scattering. *Nat. Commun.* **11**, 3889 (2020).
11. Yu, P. et al. Earth abundant materials beyond transition metal dichalcogenides: A focus on electrocatalyzing hydrogen evolution reaction. *Nano Energy* **58**, 244–276 (2019).

(5) They mention the pores of these materials and highlight them in some TEM images but they do not describe the uniformity in size or shape of these. Most of the images simply look like holes left behind from aggregation of smaller particles that leaves voids rather than pore created from a byproduct leaving.

- Many thanks. Yes, you are right. As revealed in HAADF-STEM images (**Supplementary Fig. 14**), H-Mo₂C/NG holds clear nanopores with uneven shape and size, which left behind by the gradual growth and then continuous merging together of individual domains. We have adjusted this description in the revised manuscript: “That say, with appropriate nucleation site supplying, the individual domains gradually grow and then continuously merge together on the NG substrate, which introduces abundant nanopores with uneven shape and size into polycrystalline Mo₂C NSs, achieving higher exposure of active sites for electrocatalysis.” (Page 12)

5. Is the methodology sound? Does the work meet the expected standards in your field?

Grain boundary analysis is difficult to quantify and prove. Even within the same image (ex supporting figure 6) there are several different particle sizes and grain boundaries present. I have a hard time telling the difference between the different samples. This make the data in supporting figure 8 and the major theme of this paper difficult to prove.

- We really appreciate your comments. Yes, grain boundary analysis is difficult to quantify and prove. In this work, the grain boundary density is calculated by taking the total grain boundary length and dividing by the total sample area, which can provide an accurate measurement of the relative grain boundary surface density (1-3). In our collection, the corresponding data are obtained from TEM and IPF images.

Indeed, **Supplementary Fig. 6** shows several different particle sizes and grain boundaries present in the same image, suggesting that each grain in the polycrystalline microstructure is random. For easier identification and extraction of grain boundary density information, the IPF images from PED analysis corresponding to **Supplementary Fig. 6** is shown in **Fig. 2e** and **Supplementary Fig. 7**, where the grain boundary densities for the three samples show clear differences. Furthermore, we calculated the grain boundary densities of Mo₂C nanosheets in hybrids prepared with different water contents and summarized in **Supplementary Fig. 8**, where the statistical region of each sample is ~ 2,0000 nm². Although the quantitative measurement of grain boundary density in this work is not sufficiently accurate, it does not affect the conclusion on the correlation between water content and grain boundary density. As the TEM images intuitively illustrate, with increasing water content, the structure of Mo₂C evolves from discontinuous nanosheets (H-Mo₂C/NG-7.14) (**Supplementary Fig. 16**) to continuous nanosheets with abundant grain boundaries (H-Mo₂C/NG) (**Supplementary Fig. 17**), and finally to layered stacked sheets (H-Mo₂C/NG-43.28) (**Supplementary Fig. 18**).

References

1. Heo, Y. et al. Symmetry-broken atom configurations at grain boundaries and oxygen evolution electrocatalysis in perovskite oxides. *Adv. Energy Mater.* **8**, 1802481 (2018).
2. Verdaguier-Casadevall, A. et al. Probing the active surface sites for CO reduction on oxide-derived copper electrocatalysts. *J. Am. Chem. Soc.* **137**, 9808–9811 (2015).
3. Mariano, R. G., McKelvey, K., White, H. S. & Kanan, M. W. Selective increase in CO₂ electroreduction activity at grain-boundary surface terminations. *Science* **358**, 1187–1192 (2017).

6. Is there enough detail provided in the methods for the work to be reproduced?

Yes

➤ We thank the reviewer for this very positive comments.

Reviewer #3 (Remarks to the Author):

This work by Yang and co-authors reports on electrocatalytic activity of a modified Mo₂C nanosheet for hydrogen evolution reaction (HER). They used grain boundary (GB) to tune the HER activity of the catalyst. The engineered catalyst shows a very impressive HER activity and stability in alkaline and acidic electrolytes—outperforming the benchmark Pt/C catalyst. Some insights into the mechanism of the enhanced HER performance is provided. This work is very interesting and suitable for publication in *Nat. Commun.* after the following comments are addressed.

➤ We thank the reviewer for this very positive review. The main points have been largely corrected in our revised version, and all changes are highlighted in blue throughout the revision. We wish the revision fulfills your requirements to be published in *Nature communications*.

1. The turn over frequency in KOH is relatively low. Can the authors explain the reason behind these lower values compared to those from the acidic electrolyte? I expect the TOF to be lower in basic electrolyte, but not like this.

- We really appreciate your comments. TOF is defined as the number of hydrogen molecules evolved on an active site in unit time period and is the best figure of merit to compare the catalytic activity. Typically, three elementary reactions are involved in the HER in acidic electrolytes. First, a hydrated proton associates an electron and chemically attaches on the catalyst (Volmer reaction). After that, the adsorbed H tends to form H₂ via the Heyrovsky or Tafel route, and sequentially desorbs. Likewise, the HER pathway could also be through the Volmer–Heyrovsky process or Volmer–Tafel pathways in alkaline media. Unlike the acid condition, electrochemical reduction of H₂O into adsorbed OH⁻ and adsorbed H take place in the Volmer process. The catalyst needs to break the H–O–H bond in alkaline solutions before adsorbing H, which is more difficult to pursue than the reduction of H₃O⁺. Thus, the acidic electrolytes are favorable for water electrolysis as there are enough H⁺ in the solution to react on the electrode surface. In alkaline solutions, the HER rate is 2–3 orders of magnitude lower than that in acidic solutions. Therefore, the TOF values in 1.0 M KOH observed to be lower than those in 0.5 M H₂SO₄ in this manuscript.

Assuming the cathodic current is entirely attributed to the HER, the TOF can be calculated from the j (e.g., in A cm⁻²) and the number of active sites (m) in per unit area (e.g., in cm²) by using an equation of $\text{TOF} = j/2mF$. The m is determined by cyclic voltammetry measurements between -0.2 V to 0.6 V vs. RHE in 1.0 M phosphate buffer solution (PBS, pH = 7) with a scan rate of 50 mV s⁻¹. While it is difficult to assign the observed peaks to a given redox couple, m should be proportional to the integrated charge over the whole potential range. Thus, the method of calculating TOF used in this manuscript is feasible. Meanwhile, based on the high reproducibility of polarization curves in alkaline electrolytes, the TOF value of 3.81 H₂ s⁻¹ at $\eta = 100$ mV for H-Mo₂C/NG is reasonable. In this manuscript, experiments (**Fig. 2e, d, f** and **Fig. 3**) and theoretical calculations (**Fig. 6e** and **Supplementary Figs. 53, 54**) demonstrate that the ultrathin Mo₂C nanosheets in H-Mo₂C/NG possess abundant grain boundaries, and the grain boundaries hold a large number of sites with the free energies of H adsorption (such as -0.005, -0.010, -0.013, -0.044, 0.035, and 0.038 eV) close to or even better than that of Pt (111) (-0.121 eV), indicating that the H-Mo₂C/NG catalyst affords the hydrogen production rate close to or even surpassing that of noble metal catalysts.

2. The Tafel slope in alkaline electrolyte decreases with catalyst wt% and increases after. This is different from the behavior in acidic electrolyte. Is there a relationship between this trend and the mechanism of the HER?

- We really appreciate your comments. The HER is a multi-step electrochemical process occurring on the surface of solid electrode. Typically, HER in acid and alkaline solutions involves in three principal steps: i) Volmer reaction (Tafel slope of about 120 mV dec⁻¹), ii) Heyrovsky reaction (Tafel slope of about 40 mV dec⁻¹), and iii) Tafel reaction (Tafel slope of about 30 mV dec⁻¹) (I). In both acidic and alkaline electrolytes, Tafel slopes on Mo₂C hybrids first decreased and then increased with increasing water content (**Supplementary Figs. 35, 37**). Tafel slopes of H-Mo₂C/NG are 38 and 48 mV dec⁻¹ in acidic and alkaline electrolytes, respectively, indicating the same electrocatalytic hydrogen evolution mechanism with the Heyrovsky reaction as the rate-determining step. All the above pathways intensively rely on the inherent chemical and electronic properties of the electrode surface. **In this system, the evolution trend of the Tafel slope can be attributed to the dominance of different active grain boundaries in Mo₂C hybrids obtained at varying water concentrations.** H-Mo₂C/NG is harvested when the water content of the MoO₃/RGO intermediate is 18.56 wt%, and DFT calculations reveal that the sites at fcc/hcp, fcc/fcc, and hcp/hcp grain boundaries in H-Mo₂C/NG possess optimized free energies

of hydrogen adsorption (such as -0.005 , -0.010 , -0.013 , -0.044 , 0.035 , and 0.038 eV) (**Fig. 6e** and **Supplementary Figs. 53, 54**). When the water content is below 18.56 wt%, a few grain boundaries are formed (**Supplementary Figs. 16, 17a1-a4**), and the planar and edge sites of Mo₂C grains rather than grain boundaries are the dominant active sites for catalytic hydrogen production. In acidic electrolyte, the Tafel slopes of Mo₂C/NG (137 mV dec⁻¹), H-Mo₂C/NG-0.41 (132 mV dec⁻¹), and H-Mo₂C/NG-7.14 (99 mV dec⁻¹) obtained in this system are close to that of Mo₂C particles (101.7 mV dec⁻¹) and Mo₂C sheets (144.4 mV dec⁻¹) (2) reported previously. When the water content is higher than 18.56 wt%, the Tafel slopes of the Mo₂C hybrids increases, which can be attributed to the severe agglomeration of the Mo₂C nanosheets (**Supplementary Figs. 17a1-c1, 18**), reducing the exposure of highly active sites at the grain boundaries. We have corrected the description in the revised manuscript: “Obviously, as the water content increases, the η_{10} and Tafel slopes on Mo₂C hybrids firstly decrease and then increase (**Supplementary Figs. 34, 35**), where H-Mo₂C/NG achieves the best HER activity with water content of 18.56 wt%.” (Pages 26-27)

References

1. Wang, J., Xu, F., Jin, H., Chen, Y. & Wang, Y. Non-noble metal-based carbon composites in hydrogen evolution reaction: Fundamentals to applications. *Adv. Mater.* **29**, 1605838 (2017).
2. Jia, J. et al. Ultrathin N-doped Mo₂C nanosheets with exposed active sites as efficient electrocatalyst for hydrogen evolution reactions. *ACS Nano* **11**, 12509–12518 (2017).

3. The trends in onsets and overpotentials in both electrolytic solutions are different from those observed in the Tafel analyses. Have the authors further investigated the kinetics of the electrocatalyst? Are there kinetics' limitations with the electrocatalysts wt%?

- Thank you for useful comments to improve the quality of this paper. Yes, with the increase of water content, the trends in onsets and overpotentials in alkaline electrolyte are slightly different from those observed in the Tafel analyses, which may be attributed to the different dominant active sites in the prepared Mo₂C hybrids. When the water content is low, a few grain boundaries are formed (**Supplementary Figs. 16, 17a1-a4**), and the approximative Mo₂C grain surface sites rather than the sites at the grain boundaries are the dominant sites for catalytic hydrogen production. When the water content is higher than the optimal concentration (18.56 wt%), the severe agglomeration of Mo₂C nanosheets deteriorates the species of dominant active sites at the restrictedly exposed grain boundaries (**Supplementary Figs. 17a1-c1, 18**).

Furthermore, HER kinetics have also been analyzed based on exchange current density (j_0). As water content increases, j_0 values on Mo₂C hybrids firstly increase and then decrease (**Supplementary Table 5**), where H-Mo₂C/NG achieves the highest j_0 of 5.47 and 0.75 mA cm⁻² in acidic and alkaline electrolytes, respectively. Meanwhile, H-Mo₂C/NG also exhibits lower Tafel slopes and charge transfer resistances than other control samples (**Supplementary Figs. 35, 37, 42**), which indicates that H-Mo₂C/NG possesses better HER kinetics. Compared with acidic media, the exchange current densities are lower in alkaline media, which could be attributed to the additional dissociation energy barriers of water on the electrocatalyst surfaces. We have added this discussion in the revised manuscript: “The exchange current density (j_0) of H-Mo₂C/NG is calculated to be 5.47 mA cm⁻², which is much higher than that of Mo₂C/NG (0.80 mA cm⁻²) (**Supplementary Table 5**). Distinctly, H-Mo₂C/NG with rich GBs has such a lower Tafel slope and larger j_0 value than those of Mo₂C/NG, indicating the superior HER kinetics and enhancing

the intrinsic activity.” (Page 26) “Compared to Mo₂C/NG catalyst, H-Mo₂C/NG also affords a lower Tafel slope of 48 mV dec⁻¹ (Fig. 5d) and higher j_0 value (0.70 mA cm⁻²) (Supplementary Table 5), which further underscores favorable HER kinetics and illustrates the same catalytic mechanism as in acidic media.” (Page 27)

Additionally, for H-Mo₂C/NG system, a single value for the Tafel slope is observed between -9.02 and -31.40 mV in 0.5 M H₂SO₄ and between -17.36 and -28.69 mV in 1.0 M KOH (Fig. 5b, d), indicating that the HER on the electrode is a purely kinetically limited reaction described by the Tafel equation (1, 2). And the behavior is not linear at higher overpotentials ($E > 31.40$ mV in 0.5 M H₂SO₄ and $E > 28.69$ mV in 1.0 M KOH), which indicates that the HER is not kinetically controlled. For Mo₂C/NG system, the overpotentials ranging from -130.21 to -190.97 mV and -49.90 and -81.49 mV in acidic and alkaline electrolytes are purely kinetically limited, respectively.

References

1. Herraiz-Cardona, I., González-Buch, C., Valero-Vidal, C., Ortega, E. & Pérez-Herranz, V. Co-modification of Ni-based type Raney electrodeposits for hydrogen evolution reaction in alkaline media. *J. Power Sources* **240**, 698e704 (2013).
2. Southampton Electrochemistry Group, Instrumental methods in electrochemistry. Wiley, New York, 1985.

4. The authors mentioned that Mo₂C derived from ternary compounds with termination groups OH show sluggish kinetics towards HER, but this is not necessarily true (<https://doi.org/10.1021/acsenergylett.6b00247>). Please revise this.

➤ Thank you for useful comments to improve the quality of this paper. We apologize for this inaccurate expression. We have corrected the description in the revised manuscript: “in the latter, the delamination yield and defect controllability of Mo₂C sheets derived in aqueous fluoride-containing acidic solutions are not satisfactory.” (Page 4)

5. The variation of GB with the water content is interesting. Are there any correlations between this behavior and the electrocatalytic properties? Is there an optimum GB?

➤ We really appreciate your comments. Yes, there is a certain correlation between the GBs that vary with water content and the electrocatalytic properties, that is, the higher the GB density (Supplementary Fig. 8), the better the electrocatalytic performance (Supplementary Figs. 34-37), and the dense GBs from Mo₂C nanosheets afford an ultra-high density of active sites for electrocatalytic HER. H-Mo₂C/NG achieves the best HER activity with water content of 18.56 wt%. Furthermore, DFT calculations also reveal that the grain boundaries in H-Mo₂C/NG system possess a large number of sites with the free energies of H adsorption (such as -0.005, -0.010, -0.013, -0.044, 0.035, and 0.038 eV) close to or even better than that of Pt (111) (-0.121 eV) (Fig. 6e and Supplementary Figs. 53, 54). Therefore, the abundant GBs in Mo₂C nanosheets jointly contribute to the electrocatalytic hydrogen production, and the hydrogen evolution activities of different GBs are roughly the same in this system.

6. The presence of oxides does not seem to affect the stability of the material. Have the authors confirmed the oxidation states obtained from XPS with XANES? It would be informative to look at the catalyst before and

after the reaction under XANES.

- We really appreciate your comments. As shown in **Supplementary Fig. 46**, the Mo 3d spectrum of H-Mo₂C/NG after cycling test is unambiguously deconvoluted into four pairs peaks of Mo²⁺, Mo³⁺, Mo⁴⁺, and Mo⁶⁺ species, in which the Mo²⁺ and Mo³⁺ can be ascribed to molybdenum carbides and nitrides, respectively, and the remaining Mo⁴⁺ and Mo⁶⁺ can be attributed to MoO₂ and MoO₃. Previous reports show that surface MoO₃ in Mo₂C is in situ reduced to MoO₂ species during HER (1). As indicated by DFT calculations, the free energy of hydrogen adsorption values (−0.236, −0.238, −0.260, and −0.266 eV) for reduced Mo₂C surfaces with terminal Mo=O moieties are closer to thermodynamic neutrality, but these values are not comparable to those at grain boundaries (such as −0.005, −0.010, −0.013, −0.044, 0.035, and 0.038 eV) in Mo₂C nanosheets for our H-Mo₂C/NG system (**Fig. 6e** and **Supplementary Figs. 53, 54**). Meanwhile, MoO₂ and MoO₃ are not efficient catalysts for HER (2, 3). Moreover, due to the excellent corrosion resistance of molybdenum nitride and the extremely difficult reduction of dominant Mo₂C during the HER process (1, 4, 5). Based on the above, H-Mo₂C/NG catalyst containing nitride and oxides in Mo₂C nanosheets with dense grain boundaries has excellent stability in electrocatalytic hydrogen production. As revealed in **Supplementary Fig. 43**, after 10,000 or even 50,000 cycles in 0.5 M H₂SO₄ and 1.0 M KOH, the shifts of polarization curves for H-Mo₂C/NG are negligible compared with the initial curves. Meanwhile, after 100,000 s of operation at $j = -100, -200, \text{ and } -300 \text{ mA cm}^{-2}$, the potentials of H-Mo₂C/NG only increase by 8, 14, and 38 mV in 0.5 M H₂SO₄, and 11, 13, and 36 mV in 1.0 M KOH, respectively (**Fig. 5g**). Furthermore, TEM characterization shows that Mo₂C nanocrystals in H-Mo₂C/NG still maintain an ultrathin nanosheet structure with highly exposed high-density grain boundaries after electrochemical cycling (**Supplementary Fig. 44**). Besides, XRD characterization displays that Mo₂C is well crystallized, and no new phase is experimentally detected (**Supplementary Fig. 45**). Therefore, combined with HER performance test, TEM, XRD, and XPS results after cycling have fully confirmed the excellent stability of the structure, oxidation state, and performance of the H-Mo₂C/NG catalyst during HER process, so we believe that XANES measurement after cycling are not necessary to examine the information at catalyst after the reaction.

References

1. He, L. et al. Molybdenum carbide-oxide heterostructures: In situ surface reconfiguration toward efficient electrocatalytic hydrogen evolution. *Angew. Chem. Int. Ed.* **59**, 3544–3548 (2020).
2. Li, J.-S. et al. Coupled molybdenum carbide and reduced graphene oxide electrocatalysts for efficient hydrogen evolution. *Nat. Commun.* **7**, 11204 (2016).
3. Vrubel, H. & Hu, X. Molybdenum boride and carbide catalyze hydrogen evolution in both acidic and basic solutions. *Angew. Chem. Int. Ed.* **51**, 12703–12706 (2012).
4. Zhu, J. et al. Recent advances in electrocatalytic hydrogen evolution using nanoparticles. *Chem. Rev.* **120**, 851–918 (2019).
5. Guan, H. et al. Low temperature synthesis of plasmonic molybdenum nitride nanosheets for surface enhanced Raman scattering. *Nat. Commun.* **11**, 3889 (2020).
7. The lateral layer is relatively short compared to common nanosheets, especially those obtained from chemical exfoliation. Do the authors have a control over this? I suppose the active sites are located on the basal planes. Thus, large lateral size will lead to more catalytic activity.

- We thank the reviewer for the professional question. The Mo₂C in our work presents an ultrathin nanosheet morphology with several micrometers in lateral size, which is relatively short compared to common nanosheets, especially those obtained from chemical exfoliation. And we have considered the size control of the lateral layer. **The lateral sizes of Mo₂C nanosheets depend on the water content of the MoO₃ intermediate and the oxidation degree of graphene oxide.** Firstly, the lateral size and grain boundary density of Mo₂C nanosheets are sensitive to water. With the increase of water content, the structure of Mo₂C evolves from nanoparticles (Mo₂C/NG) (**Supplementary Fig. 2**) to discontinuous nanosheets (H-Mo₂C/NG-7.14) (**Supplementary Fig. 16**), then to continuous nanosheets with rich grain boundaries (H-Mo₂C/NG) (**Supplementary Fig. 17**), and finally to layered stacked sheets (H-Mo₂C/NG-43.28) (**Supplementary Fig. 18**). The grain boundary density of Mo₂C nanosheets in Mo₂C hybrids is summarized in **Supplementary Fig. 8**. Obviously, H-Mo₂C/NG possesses the highest grain boundary density of $138 \pm 3 \mu\text{m}^{-1}$, distinctly proving that the water content in MoO₃/RGO effectively manipulates the grain boundary density of Mo₂C nanosheets. Meanwhile, sufficient oxygen-containing functional groups on the graphene oxide surface are necessary for the formation of high-density GBs in Mo₂C NSs. Specifically, due to the limited nucleation sites of graphene oxide with low oxidation degree, Mo₂C nanosheets with a small amount grain boundaries are obtained (**Supplementary Fig. 11**). While the morphology of Mo₂C product supported with excessively oxidized graphene oxide is slightly agglomerated (**Supplementary Fig. 12**) as compared with the uniformly structure of H-Mo₂C/NG synthesized with common graphene oxide as substrate (**Fig. 2d**). During the carburization process, the escape of a large number of oxygen-containing functional groups on graphene oxide severely damages the framework structure, which inhibits the lateral size of Mo₂C nanosheets, leading to a decrease in the number of active sites and activity. **Thus, when common graphene oxide is used as the substrate and the water content of the MoO₃/RGO intermediate is 18.56 wt%, structurally optimized Mo₂C nanosheets with a lateral size of several micrometers and fully exposed grain boundaries are obtained.**

8. The correlation between the Mo orbital energy level and H_{ads} is interesting. However, more fundamental insights into the mechanism of HER will be useful. Have the authors explored this using DFT calculations? These calculations must be done in aqueous environment to take into account the solvation effect.

- Thanks to the reviewer for the high evaluation. Indeed, this work experimentally confirms that water induces the formation of Mo₂C nanosheets with abundant grain boundaries during carbonization process. Theoretical calculations propose the correlation between the Mo d_{z²} orbital energy level and H_{ads}. The variation in the hydrogen–metal bond depends, to a large extent, on the strength of the coupling between the hydrogen s states and the metal d states. This coupling forms bonding and antibonding states as illustrated in **Fig. 6d**. This descriptor role of H_{ads} gives a hint that HER could be described by metal d orbital energy level, in the conventional point of view, which is a rational reflection of hydrogen binding energy. In this work, theoretical calculation results suggest that the H_{ads} values (such as -0.005, -0.010, -0.013, -0.044, 0.035, and 0.038 eV) at the grain boundary regions are closest to the optimal energy around 0 eV, and the H_{ads} values for the sites away from the grain boundary regions are well below the 0 eV (such as -0.324, -0.358, -0.428, and -0.517 eV) (**Fig. 6e** and **Supplementary Figs. 53, 54**). These results indicate the domain interfaces of Mo₂C nanosheets have much more activity than the planar crystal. Particularly, the Mo d_{z²} orbital energy level exhibits an intrinsic relationship with the electrocatalyst activity and is regarded as a new descriptor for accurately predicting the HER activity.

Our finding broadens the applicability of d band theory to activity prediction of metal electrocatalysts. We have added a detailed description of DFT calculations in the revised manuscript: “The strength of H atom binding to the surface was determined by the bond strength of the H–Mo bond, which was determined by the orbital overlap between the H and surface Mo atoms. The classical molecular orbital overlap theory was used to explain the overlap between the d_z^2 orbital of Mo and s orbital of H. Moreover, the detailed electronic structures of Mo_2C hybrids were demonstrated by PDOS with the DFT calculation.” (Page 36)

For the environment of H adsorption, this is a good question. We have done the calculation of H_{ads} in the water solution environment by the vasp-sol method following the reviewer’s suggestion. And no obvious changes have been found between the current vacuum model and solution mode (**Table 1**). The calculation detail:

We use the (002)/(100) interface of Mo_2C (**Figure 1**) as the model for the calculation of the free energy of hydrogen adsorption in the water solution environment, as shown in **Figure 2**.

Figure 1. Atomic structure model of Mo_2C (002)/(100) interface in the vacuum environment.

Figure 2. Atomic structural model of Mo_2C (002)/(100) interface in the water solution environment. (a-d) Configure 1-4.

Table 1. The calculation of the free energy of hydrogen adsorption in the vacuum and water solution environments.

Environments	H_{ads_vacuum} (eV)	H_{ads_solution} (eV)
Configure 1	0.08953605	0.12893605
Configure 2	-0.7270639	-0.742264
Configure 3	-0.9825639	-0.9604639
Configure 4	-0.6284639	-0.593864

Reviewers' comments:

Reviewer #1 (Remarks to the Author):

In the current work, the authors answered all questions for reviewers. After reorganizing the manuscript, the authors still did not persuade readers to understand the reason why there is only grain boundary affects the Mo₂C/NG HER performance.

1. There is a polycrystalline material, there are different proportions of exposed crystal planes. There is not only one crystal plane, do different crystal planes affect the HER performance? Do they need a compare experiment for clarifying the single crystal plane HER performance with or without grain boundary?
2. In a water-assisted system, Mo₂C nanosheet grows in different thicknesses, then it is a different contact area with electrolyte. How about the Mo₂C performance of other thicknesses?
3. In SI fig.14, the authors re-said that with appropriate nucleation site supplying, the individual domains gradually grow and then continuously merge on the NG substrate, which introduces abundant nanopores with uneven shape and size into polycrystalline Mo₂C NSs, achieving higher exposure of active sites for electrocatalysis. How do we distinguish the electrocatalytic contributions from grain bounds or nanopores?

Reviewer #3 (Remarks to the Author):

The authors have addressed my comments.

Reviewer #4 (Remarks to the Author):

The authors propose a synthesis method to obtain Mo₂C nanosheets with a high density of grain boundaries. They claim that such material can achieve hydrogen evolution reaction activity similar to what is observed for Pt. The research topic is highly relevant, and the work contains a substantial amount of data and nice results that can be interesting for other researchers in the field.

Similar materials have been highly investigated for the same reaction considering different structures and doping strategies, and they often indicate highly efficient catalyst for HER [Nature Comm., 12, 6776 (2021); Chem. Sci., 7, 3399-3405 (2016); ACS Nano, 11, 12, 12509-12518 (2017)]. The novelty of this work is that the inclusion of water in the synthesis method could lead to Mo₂C NS with more GBs, the attempt to explain the mechanism that generates this effect, and the connection between the GBs sites and the high catalytic activity. However, there are still some severe deficiencies in the manuscript, and I cannot recommend it for publication in Nature Communications in its current form.

1) The description of the theoretical methodology is far from complete, which would make it difficult for others to reproduce the work. For instance, the authors don't provide details about electronic and ionic convergence criteria, possible supercell sizes, k-point mesh, usage of dipole corrections, or solvation corrections.

2) The authors propose a mechanism using experiments and DFT calculations to explain the impact of water in the generation of different morphologies of Mo₂C. The discussion around Figure 1b using DFT is based on the two theoretical models of MoO₃ with and without water. However, the text lacks details about the atomic models and the reasons for each assumption in the considered mechanism towards Mo₂C structures. For instance, how are the authors modeling the MoO₃ with and without

water? What are the assumptions to generate the TSNP, TSNS, TSNP-C, TSNS-C, TSNP-Ph, TSNS-Ph models from the ISNP and ISNS? How is the reaction enthalpy calculated for each considered step?

3) How did the authors construct computational models of fcc/hcp, fcc/fcc, and hcp/hcp Mo₂C GBs?

4) The authors claim that “Hads on the Mo atoms near the fcc/fcc and hcp/hcp homophase Mo₂C GBs also display close-to-zero values, such as -0.044, 0.035, and 0.038 eV”. However, Supplementary Figs. 54 shows Hads with values lower than -0.75 and higher than 0.75 eV in some cases. The results from Fig 6 e) and Supplementary Figs. 54 do not indicate which adsorption site represents each point. Thus, the reader will not be able to distinguish active and inactive regions in the simulated fcc/fcc and hcp/hcp Mo₂C GBs using the reported data.

5) How were the adsorption sites of the models represented in Supplementary Figs. 54 sampled?

6) In addition to the Hads close to zero, other factors such as coverage effects and kinetic limitations that are not directly included in the model can contribute to the observed catalytic activity. [ACS Catal., 10, 121–128 (2020)] Further comments on such variables that are ignored in the current model could benefit the manuscript discussion.

7) About the usage of the dz₂ orbital energy level as a descriptor for Hads. What is the error between the proposed fitting equation and the calculated Hads? Are all the points from Supplementary Figs. 54 used to generate Fig 6 e)?

8) In different parts of the text, the authors claim to have found a new and improved descriptor for HER. “Compared with the state-of-the-art descriptors such as band-energy theory and work function that describe the properties of the whole surface, the dz₂ orbital energy level is more relevant to the active site of the catalysts.”. However, the descriptor is tested only for one type of material. More data with different materials would be necessary to support that this descriptor is transferable enough to “broaden” the applicability of the d-band theory.

9) Please provide the atomic coordinates of the calculations (CONTCARS), so interested readers can reproduce the work.

Response to the reviewers

Reviewer #1 (Remarks to the Author):

In the current work, the authors answered all questions for reviewers. After reorganizing the manuscript, the authors still did not persuade readers to understand the reason why there is only grain boundary affects the Mo₂C/NG HER performance.

➤ We are grateful for the reviewer's professional comments and suggestions. We are sorry that there are lots of questions that we are not explained clear in the previous version. In particular, the effect of grain boundaries on the HER performance of catalysts. In this system, the grain boundaries in Mo₂C nanosheets are proposed to be the dominant factor, rather than the sole factor, affecting the HER performance of H-Mo₂C/NG. We have replied to your comments that are given below. We wish the revision fulfills your requirements to be published in *Nature Communications*.

1. There is a polycrystalline material, there are different proportions of exposed crystal planes. There is not only one crystal plane, do different crystal planes affect the HER performance? Do they need a compare experiment for clarifying the single crystal plane HER performance with or without grain boundary?

➤ We really appreciate your comments. Yes, in polycrystalline Mo₂C material, different crystal planes can affect the HER performance. Previous reports show that the free energies of H adsorption at the widely studied Mo₂C (001), (100), (110), and (101) as dominant HER activity are -0.87 (1), -0.63 , -0.59 (2), and -0.33 (3), respectively, which are far below the optimal value (close-to-zero), indicating raw Mo₂C nanocrystals have inactive HER activity, consistent with the experimental results (4, 5). In our work, DFT calculations show optimized H adsorption free energies (such as -0.005 , -0.010 , -0.013 , -0.044 , 0.035 , and 0.038 eV) at the fcc/hcp, fcc/fcc, and hcp/hcp grain boundaries (**Fig. 6e** and **Supplementary Figs. 53, 54**), which are superior to those of sites far from the grain boundaries (such as -0.324 , -0.358 , -0.428 , and -0.517 eV). Meanwhile, experimentally, TEM and IPF images intuitively show that the grain boundary density is very sensitive to water. With the increase of water content, the structure of Mo₂C evolves from nanoparticles (Mo₂C/NG) (**Supplementary Fig. 2**) to discontinuous nanosheets (H-Mo₂C/NG-7.14) (**Supplementary Fig. 16**), then to continuous nanosheets with rich grain boundaries (H-Mo₂C/NG) (**Supplementary Fig. 17**), and finally to layered stacked sheets (H-Mo₂C/NG-43.28) (**Supplementary Fig. 18**). The grain boundary density of Mo₂C nanosheets in Mo₂C hybrids is summarized in **Supplementary Fig. 8**, where the statistical region of each sample is $\sim 2,0000$ nm² and H-Mo₂C/NG with water content of 18.56 wt% possesses the highest grain boundary density of 138 ± 3 μm^{-1} . Particularly, as the water content increases, the η_{10} and Tafel slopes on Mo₂C hybrids firstly decrease and then increase in acidic (**Supplementary Figs. 34, 35**) and alkaline

(**Supplementary Figs. 36, 37**) electrolytes, where H-Mo₂C/NG with water content of 18.56 wt% achieves the best HER activity. Therefore, combined with previous reports and our theoretical and experimental results have fully confirmed that the dominant active sites for HER are located at the grain boundaries of Mo₂C nanosheets, and the high-density grain boundaries effectively alter the atomic and electronic structure of Mo₂C NSs, which provides an ultra-high fraction of active sites for HER, significantly improving the intrinsic catalytic. Thus, we believe that compare experiment for clarifying the single crystal plane HER performance with or without grain boundary are not necessary.

References

1. Han, W. et al. Ultra-small Mo₂C nanodots encapsulated in nitrogen-doped porous carbon for pH-universal hydrogen evolution: Insights into the synergistic enhancement of HER activity by nitrogen doping and structural defects. *J. Mater. Chem. A* **7**, 4734 (2019).
2. Yuan, W. et al. Two-dimensional lamellar Mo₂C for electrochemical hydrogen production: Insights into the origin of hydrogen evolution reaction activity in acidic and alkaline electrolytes. *ACS Appl. Mater. Interfaces* **10**, 40500–40508 (2018).
3. He, L. et al. Molybdenum carbide-oxide heterostructures: In situ surface reconfiguration toward efficient electrocatalytic hydrogen evolution. *Angew. Chem. Int. Ed.* **59**, 3544–3548 (2020).
4. Ma, Y. et al. Synergistically tuning electronic structure of porous β -Mo₂C spheres by Co doping and Mo-vacancies defect engineering for optimizing hydrogen evolution reaction activity. *Adv. Funct. Mater.* 2000561 (2020).
5. Wan, C., Regmi Y. N. & Leonard B. M. Multiple phases of molybdenum carbide as electrocatalysts for the hydrogen evolution reaction. *Angew. Chem. Int. Ed.* **53**, 6407–6410 (2014).

2. In a water-assisted system, Mo₂C nanosheet grows in different thicknesses, then it is a different contact area with electrolyte. How about the Mo₂C performance of other thicknesses?

- We really appreciate your comments. Yes, in the water-assisted system, the grown Mo₂C nanosheet has different thicknesses, then it accordingly gets a different contact area with electrolyte. Specifically, the radial thickness of Mo₂C nanosheets decreases with increasing lateral size as the loading of Mo₂C remains constant. Experimentally, AFM, TEM, and IPF images intuitively show that the lateral size of Mo₂C nanosheets and grain boundary density are very sensitive to water (**Fig. 2** and **Supplementary Fig. 7**). With the increase of water content, the structure of Mo₂C evolves from nanoparticles (~ 5 nm diameter) (Mo₂C/NG) (**Fig. 2a** and **Supplementary Fig. 2**) to discontinuous nanosheets (H-Mo₂C/NG-7.14) (**Supplementary Fig. 16**), then to continuous nanosheets with rich grain boundaries (H-Mo₂C/NG) (~ 1.0 nm thickness) (**Fig. 2c** and **Supplementary Fig. 17**), and finally to layered stacked sheets (H-Mo₂C/NG-43.28) (**Supplementary**

Fig. 18). Theoretically, with the increase of water content, the thinner of Mo₂C nanosheets, the larger of the lateral size and the higher of the grain boundary density. Actually, with the increase of water content, both the BET area and pore size of Mo₂C hybrids increase initially and then decrease afterwards (**Supplementary Figs. 19, 20**), indicating that the contact area between Mo₂C nanosheets and electrolytes also increases first and then decreases during the HER process. The decrease in BET area of Mo₂C hybrids is attributed to the selectively etching effect of water on graphene framework during carbonization process. Noteworthy, the exposed grain boundary density of Mo₂C NSs in Mo₂C hybrids increase initially and then decrease afterwards with the increase of water content (**Supplementary Fig. 8**), H-Mo₂C/NG with water content of 18.56 wt% possesses the highest grain boundary density of $138 \pm 3 \mu\text{m}^{-1}$. Particularly, as the water content increases, the η_{10} and Tafel slopes on Mo₂C hybrids firstly decrease and then increase in acidic (**Supplementary Figs. 34, 35**) and alkaline (**Supplementary Figs. 36, 37**) electrolytes, where H-Mo₂C/NG achieves the best HER activity. Thus, the grain boundary has a certain correlation with the electrocatalytic HER performance; that is, the higher of the grain boundary density, the better of the electrocatalytic performance. Furthermore, DFT calculations also show optimal H adsorption free energies (such as -0.005, -0.010, -0.013, -0.044, 0.035, and 0.038 eV) at the fcc/hcp, fcc/fcc, and hcp/hcp grain boundaries (**Fig. 6e** and **Supplementary Figs. 53, 54**), which are superior to those of sites far from the grain boundaries (such as -0.324, -0.358, -0.428, and -0.517 eV). Therefore, the abundant grain boundaries in Mo₂C nanosheets provide the dominant active sites for HER, while the influence of thickness on performance is negligible compared to grain boundary effects in surface chemical reactions.

3. In SI fig.14, the authors re-said that with appropriate nucleation site supplying, the individual domains gradually grow and then continuously merge on the NG substrate, which introduces abundant nanopores with uneven shape and size into polycrystalline Mo₂C NSs, achieving higher exposure of active sites for electrocatalysis. How do we distinguish the electrocatalytic contributions from grain bounds or nanopores?

➤ We appreciate the reviewer's professional comments. During carbonization process, with appropriate nucleation site supplying, the individual Mo₂C domains gradually grow and then continuously merge on the NG substrate, which introduces abundant nanopores with uneven shape and size into polycrystalline Mo₂C NSs, achieving high exposure of active sites for electrocatalysis. In fact, the contact surfaces with the electrolytes at the edge of the nanopores are the crystal planes of Mo₂C domains. Previous reports have revealed that raw Mo₂C nanocrystals have inactive HER activity. In detail, the free energies of H adsorption at the widely studied Mo₂C (001), (100), (110), and (101) as dominant HER activity are -0.87 (1), -0.63, -0.59 (2), and -0.33 (3), respectively, which are far below the optimal value (close-to-zero). And raw Mo₂C require extremely large overpotentials of about 331 and 230 mV at pH = 0 and 14 to reach a current density of 10 mA cm⁻²,

respectively (1, 4). In our system, H-Mo₂C/NG requires overpotentials of only 10 and 63 mV to achieve a current density of -10 mA cm^{-2} in acidic and alkaline electrolytes, respectively, which far surpasses those of raw Mo₂C (331 and 230 mV) and Mo₂C/NG (138 and 107 mV) and is even comparable to that of state-of-the-art Pt/C catalyst (19 and 57 mV) (**Fig. 5**), indicating that the grain boundaries in Mo₂C nanosheets effectively reduce the energy input for activating HER. Meanwhile, DFT calculations also show optimal H adsorption free energies (such as -0.005 , -0.010 , -0.013 , -0.044 , 0.035 , and 0.038 eV) at the fcc/hcp, fcc/fcc, and hcp/hcp grain boundaries (**Fig. 6e** and **Supplementary Figs. 53, 54**), which are superior to those of sites far from the grain boundaries (such as -0.324 , -0.358 , -0.428 , and -0.517 eV). Therefore, combining with previous reports and our theoretical and experimental results confirm that the nanopores contribute negligibly to the intrinsic HER activity in this work. The abundant grain boundaries in Mo₂C nanosheets provide the dominant active sites for HER, and the nanopores facilitate the mass transfer during the HER process, thereby accelerating the hydrogen production rate.

References

1. Han, W. et al. Ultra-small Mo₂C nanodots encapsulated in nitrogen-doped porous carbon for pH-universal hydrogen evolution: Insights into the synergistic enhancement of HER activity by nitrogen doping and structural defects. *J. Mater. Chem. A* **7**, 4734 (2019).
2. Yuan, W. et al. Two-dimensional lamellar Mo₂C for electrochemical hydrogen production: Insights into the origin of hydrogen evolution reaction activity in acidic and alkaline electrolytes. *ACS Appl. Mater. Interfaces* **10**, 40500–40508 (2018).
3. He, L. et al. Molybdenum carbide-oxide heterostructures: In situ surface reconfiguration toward efficient electrocatalytic hydrogen evolution. *Angew. Chem. Int. Ed.* **59**, 3544–3548 (2020).
4. Ma, Y. et al. Synergistically tuning electronic structure of porous β -Mo₂C spheres by Co doping and Mo-vacancies defect engineering for optimizing hydrogen evolution reaction activity. *Adv. Funct. Mater.* 2000561 (2020).

Reviewer #4 (Remarks to the Author):

The authors propose a synthesis method to obtain Mo₂C nanosheets with a high density of grain boundaries. They claim that such material can achieve hydrogen evolution reaction activity similar to what is observed for Pt. The research topic is highly relevant, and the work contains a substantial amount of data and nice results that can be interesting for other researchers in the field.

Similar materials have been highly investigated for the same reaction considering different structures and doping strategies, and they often indicate highly efficient catalyst for HER [Nature Comm., 12, 6776 (2021); Chem. Sci., 7, 3399–3405 (2016); ACS Nano, 11, 12, 12509–12518 (2017)]. The novelty of this work is that the inclusion

of water in the synthesis method could lead to Mo₂C NS with more GBs, the attempt to explain the mechanism that generates this effect, and the connection between the GBs sites and the high catalytic activity. However, there are still some severe deficiencies in the manuscript, and I cannot recommend it for publication in Nature Communications in its current form.

➤ We sincerely thank the reviewer for his/her valuable comments and acknowledge the importance of this manuscript. We are sorry that there are some severe deficiencies in the previous version. We have improved the manuscript (all changes highlighted in blue throughout the revision) and the replies to your comments are given below. We wish the revision fulfills your requirements to be published in *Nature Communications*.

1. The description of the theoretical methodology is far from complete, which would make it difficult for others to reproduce the work. For instance, the authors don't provide details about electronic and ionic convergence criteria, possible supercell sizes, k-point mesh, usage of dipole corrections, or solvation corrections.

➤ We thank the reviewer for the suggestion. We have added the calculation detail as the reviewer's comments in the revised supporting information: "During the simulation lattice parameters were fully relaxed, geometry of crystal was fully optimized on the basis of the criteria of a maximum atomic force of 0.01 eV Å⁻¹ and energy convergence criteria of 0.1 meV before the electron structure and total energy calculation. The VASPSOL³⁶ function was employed to mimic the solvation environment. 15 Å vacuum space and dipole correction were added to the 2D slab model to remove the convergence problem. Different supercells (2x2x1 to 4x4x1) were adopted during the calculation and the k-mesh point convergence test was conducted for each supercell." (Page 68)

Reference

36. Mathew, K., Sundararaman, R., Letchworth-Weaver, K., Arias, T. A. and Hennig, R. G. Implicit solvation model for density-functional study of nanocrystal surfaces and reaction pathways. *J. Chem. Phys.* **140**, 084106 (2014).

2. The authors propose a mechanism using experiments and DFT calculations to explain the impact of water in the generation of different morphologies of Mo₂C. The discussion around Figure 1b using DFT is based on the two theoretical models of MoO₃ with and without water. However, the text lacks details about the atomic models and the reasons for each assumption in the considered mechanism towards Mo₂C structures. For instance, how are the authors modeling the MoO₃ with and without water? What are the assumptions to generate the TSNP, TSNS, TSNP-C, TSNS-C, TSNP-Ph, TSNS-Ph models from the ISNP and ISNS? How is the reaction enthalpy calculated for each considered step?

- We thank the reviewer for the comments and suggests. Experimentally, $(\text{NH}_4)_6\text{Mo}_7\text{O}_{24}\cdot 4\text{H}_2\text{O}$ as molybdenum source is firstly added to graphene oxide (GO) dispersion to form a stable aqueous suspension due to the electrostatic repulsion of negatively charged GO and polyoxoanions. Subsequently, the integrated precursor is subjected to a hydrothermal reaction at 190 °C for 12 h to harvest MoO_3/RGO intermediate. Previous reports (1, 2) reveal that the hydrothermal reaction conditions used in this work result in the formation of MoO_3 from $(\text{NH}_4)_6\text{Mo}_7\text{O}_{24}\cdot 4\text{H}_2\text{O}$, and further we demonstrate that the generated MoO_3 is amorphous by XRD characterization (**Supplementary Fig. 1**). MoO_3/RGO intermediates with different water contents (0, 0.41, 7.14, 13.52, 18.56, 30.98, and 43.28 wt%) are obtained by freeze-drying. Meanwhile, HER performance tests find that $\text{Mo}_2\text{C}/\text{NG}$ and $\text{H-Mo}_2\text{C}/\text{NG}$ obtained from anhydrous MoO_3 and MoO_3 with water content of 18.56 wt% have the worst and best activities, respectively. Accordingly, we chose the MoO_3 structure model with water free and water content of 18.56 wt% to study the mechanism of water action in the carbothermal reaction. We searched MoO_3 structures with different water contents in the previously published literatures, and found that the structure of IS_{NS} with water content of 16% was similar to our experimental synthesized structure, and defined anhydrous MoO_3 as the IS_{NP} structure. Since MoO_3 is a typical layered structure, the layers are staggered and arranged by van der Waals force, which makes it prone to decompose at high temperature. Thus, in our case, IS_{NP} and IS_{NS} first absorb heat and dissociate into transition states of TS_{NP} and TS_{NS} , respectively, and they then react with C atoms in RGO to form Mo_2C . Generally, RGO has two types of C atoms, namely, the highly reactive carbon at edge and the relatively inertness carbon of the six-membered C-ring far away from edge (3). Because H bonded graphene edge has a structure analogous to benzene (Ph), the chemical properties of the compounds can be thought to be similar (4). For simplicity, the C atoms extracted from the edge of the graphene lattice, as well as the free Ph radicals, react with TS_{NP} and TS_{NS} to produce $\text{TS}_{\text{NP-C}}$, $\text{TS}_{\text{NP-Ph}}$, $\text{TS}_{\text{NS-C}}$, and $\text{TS}_{\text{NS-Ph}}$, respectively. Although these model structures are much simpler than those in actual experiment, they are sufficient to study the interaction between two substances. We optimize the structures of the reactants and products, and the reaction enthalpy is calculated by the enthalpy difference of products and reactants. We have added the calculation detail as the reviewer's comments in the revised supporting information: "The reaction enthalpy during Mo_2C synthesis was calculated by the enthalpy difference between the products and reactants after geometry optimization. MoO_3/RGO intermediates with different water contents (0, 0.41, 7.14, 13.52, 18.56, 30.98, and 43.28 wt%) were obtained by freeze-drying. And HER performance tests found that $\text{Mo}_2\text{C}/\text{NG}$ and $\text{H-Mo}_2\text{C}/\text{NG}$ obtained from anhydrous MoO_3 and MoO_3 with water content of 18.56 wt% had the worst and best activities, respectively (**Fig. 5**). Accordingly, we choose the MoO_3 structure model with water free and water content of 18.56 wt% to study the mechanism of water action in the carbothermal reaction. We searched MoO_3 structures with different water contents in the previously published literatures, and found that the structure of IS_{NS} with

water content of 16% was similar to our experimental synthesized structure, and defined anhydrous MoO₃ as the IS_{NP} structure. Since MoO₃ is a typical layered structure, the layers are staggered and arranged by van der Waals force, which makes it prone to decompose at high temperature. In our case, IS_{NP} and IS_{NS} first absorb heat and dissociate into transition states of TS_{NP} and TS_{NS}, respectively, and they then react with C atoms in RGO to form Mo₂C.

The reaction enthalpy was calculated as:

$$\Delta H = H_{\text{products}} - H_{\text{reactants}} \quad (5)''$$

(Page 68-69)

References

1. Chen, W. F. et al. Highly active and durable nanostructured molybdenum carbide electrocatalysts for hydrogen production. *Energy Environ. Sci.* **6**, 943–951 (2013).
2. Luo, Y. et al. A high-performance oxygen electrode for Li–O₂ batteries: Mo₂C nanoparticles grown on carbon fibers. *J. Mater. Chem. A*, **5**, 5690 (2017).
3. Bellunato, A., Tash, H. A., Cesa, Y. & Schneider, G. F. Chemistry at the edge of graphene. *ChemPhysChem* **17**, 785–801 (2016).
4. Sun, Z., James, D. K. & Tour, J. M. Graphene chemistry: Synthesis and manipulation. *J. Phys. Chem. Lett.* **2**, 2425–2432 (2011).

3. How did the authors construct computational models of fcc/hcp, fcc/fcc, and hcp/hcp Mo₂C GBs?

- We thank you for your comments. We are very sorry for not detailing the construction of the computational GBs model. Specifically, **Fig. 3e** shows the superstructure of fcc/hcp GB in Mo₂C nanosheets. Along the fcc/hcp heterophase edges, both hcp phase with “ABAB” stacking and fcc phase with “ABCABC” stacking are observed along the close-packed [110]_f/[11 $\bar{2}$ 0]_h direction, which clearly reveals that the heterophase grain boundary is generated through the chemical connection of fcc(111) and hcp(002). By analogy, according to the HAADF-STEM images of Mo₂C NSs, we use the software of VESTA to successfully construct fcc(111)/fcc(010) (**Supplementary Fig. 25**), hcp(002)/hcp(110), hcp(101)/hcp(110), hcp(010)/hcp(001), and hcp(010)/hcp(101) models (**Fig. 3a, c** and **Supplementary Fig. 25**) to simulate the fcc/hcp, fcc/fcc, and hcp/hcp Mo₂C GBs in actual material. The structural models obtained by subsequent optimization screening are shown in **Supplementary Figs. 53, 54**.

4. The authors claim that “H_{ads} on the Mo atoms near the fcc/fcc and hcp/hcp homophase Mo₂C GBs also display close-to-zero values, such as –0.044, 0.035, and 0.038 eV”. However, **Supplementary Figs. 54** shows H_{ads} with values lower than –0.75 and higher than 0.75 eV in some cases. The results from **Fig 6 e** and **Supplementary Figs. 54** do not indicate which adsorption site represents each point. Thus, the reader

will not be able to distinguish active and inactive regions in the simulated fcc/fcc and hcp/hcp Mo₂C GBs using the reported data.

- We thank you for your comments. We are sorry that many details we are not expressed clearly in the previous version. There are various H adsorption sites in the computational models of fcc/hcp, fcc/fcc, and hcp/hcp Mo₂C GBs, including the optimal active and low active sites, and we have explicitly pointed out in the revised supporting information. As revealed in **Supplementary Fig. 53**, the optimal active sites near fcc/hcp heterophase GBs are highlighted by yellow circles, the rest regions are low active. Similarly, the optimal active sites near fcc/fcc and hcp/hcp homophase GBs are highlighted by yellow circles in **Supplementary Fig. 54**, the rest regions are low active. More details are provided in revised supporting information (**Supplementary Figs. 53, 54**). (Page 57-58)

Supplementary Fig. 53. Atomic structure of fcc/hcp GB. Mo atoms with optimized H_{ads} at fcc/hcp GB are highlighted by yellow circles. The cyan and dark gray balls represent Mo and C atoms, respectively. The vicinity of the fcc/hcp GB exhibits optimal H adsorption free energies (such as -0.005 , -0.010 , -0.013 , and 0.058 eV), which are superior to those of sites far from the GBs (such as -0.324 , -0.358 , -0.428 , and -0.517 eV) (**Fig. 6e**).

Supplementary Fig. 54. Atomic structure of fcc/fcc and hcp/hcp GBs. a fcc/fcc and **b-e** hcp/hcp GBs. **f** H_{ads} diagram. The dashed line is the H_{ads} of Pt (111) (-0.121 eV). The data points in the shaded area in **f** are generated using the adsorption sites indicated by the yellow circles. The cyan and dark gray balls represent Mo and C atoms, respectively.

5. How were the adsorption sites of the models represented in Supplementary Fig. 54 sampled?

- We thank you for your comments. In **Supplementary Fig. 54**, we calculated all possible adsorption sites on the heterostructure surface and highlighted the sites with H_{ads} close to the optimization with yellow circles. We have updated the **Supplementary Fig. 54** in the revised supporting information to make it more clearly. More details are provided in revised supporting information (**Supplementary Fig. 54**). (Page 58)

Supplementary Fig. 54. Atomic structure of fcc/fcc and hcp/hcp GBs. a fcc/fcc and **b-e** hcp/hcp GBs. **f** H_{ads} diagram. The dashed line is the H_{ads} of Pt (111) (-0.121 eV). The data points in the shaded area in **f** are generated using the adsorption sites indicated by the yellow circles. The cyan and dark gray balls represent Mo and C atoms, respectively.

6. In addition to the H_{ads} close to zero, other factors such as coverage effects and kinetic limitations that are not directly included in the model can contribute to the observed catalytic activity. [ACS Catal., 10, 121–128 (2020)] Further comments on such variables that are ignored in the current model could benefit the manuscript discussion.

➤ We really appreciate your comments. In 1958, Parson first pointed out that a maximum exchange current density would be attained when the hydrogen adsorption free energy was close to thermoneutral (~ 0 eV) (1). Norskov et al. plotted hydrogen adsorption free energy from DFT calculations versus experimental HER exchange current density (2), and observed a volcano-shaped variation with the peak position close to platinum—note that platinum was the state-of-the-art HER catalyst with an almost zero overpotential and small Tafel slope. This volcano plot was further consummated in other studies in the literatures (3, 4). These reports suggest that hydrogen adsorption free energy can be exploited as an effective descriptor in the design, engineering and optimization of HER catalysts. Per Lindgren (ACS Catal., 10, 121–128 (2020)) clearly state that: “despite similar hydrogen-binding energies, both the Volmer and Tafel barriers on Au are significantly higher than on Pt. This is a consequence of the reactions proceeding from hollow sites on Au, rather than top sites on Pt. We first examine the H_2 -liberating Tafel reaction which limits the rate on Pt; this is not an electrochemical

reaction. The energetics of this reaction on Au and Pt are nearly identical; however, the barrier on Au is higher by nearly a factor of 2. The reaction from the hollow sites on Au follows a high-energy stretch, which is predominantly perpendicular to the surface, while the reaction from Pt's top sites follows lower-energy lateral bending modes." In our case, the H is adsorbed on top of the Mo atoms, which is the same as that on the surface of Pt. Therefore, we can safely claim that the applicability of the computational hydrogen electrode (CHE) has no problem. As for other factors such as coverage effect, the coverage is strongly dependent on the applied potential (Figure 4 in *ACS Catal.*, 10, 121–128 (2020)), which is not applicable to our case, where we applied very small potential. In our system, the dilute adsorption model is consistent with the experiment.

References

1. Parsons, R. The rate of electrolytic hydrogen evolution and the heat of adsorption of hydrogen. *Trans. Faraday Soc.* **54**, 1053–1063 (1958).
2. Nørskov, J. K. et al. Trends in the exchange current for hydrogen evolution. *J. Electrochem. Soc.* **152**, J23 (2005).
3. Greeley, J., Jaramillo, T. F., Bonde, J., Chorkendorff, I. B. & Nørskov, J. K. Computational high-throughput screening of electrocatalytic materials for hydrogen evolution. *Nat. Mater.* **5**, 909–913 (2006).
4. Zheng, Y. et al. Hydrogen evolution by a metal-free electrocatalyst. *Nat. Commun.* **5**, 3783 (2014).

7. About the usage of the d_z^2 orbital energy level as a descriptor for H_{ads} . What is the error between the proposed fitting equation and the calculated H_{ads} ? Are all the points from Supplementary Figs. 54 used to generate Fig 6 e?

- Thank you for useful comments to improve the quality of this paper. We apologize for the omission of the error between the proposed fitted equation and the calculated H_{ads} . We have added the error in the **Fig. 6e**. Moreover, surface Mo sites on the fcc/hcp GB model from **Supplementary Fig. 53** are used to generate **Fig. 6e**.

More details are provided in revised manuscript (**Fig. 6e**). (Page 31)

Fig. 6. Relationship between HER activity and the d_z^2 orbital energy level. a Structural models and relevant electronic states of Mo d orbital for MoC₃ configurations. S1: MoC₃ triangular plane, S2: intermediate height, S3: high height. PDOS of Mo d orbital for **b** S2 and **c** S3 structures. **d** Schematic diagram of hybridization of H 1s orbital and Mo d_z^2 orbital. **e** Correlation of the Mo d_z^2 orbital energy level and H_{ads} , inset is the corresponding fitting equation.

8. In different parts of the text, the authors claim to have found a new and improved descriptor for HER. “Compared with the state-of-the-art descriptors such as band-energy theory and work function that describe the properties of the whole surface, the d_z^2 orbital energy level is more relevant to the active site of the catalysts.”. However, the descriptor is tested only for one type of material. More data with different materials would be necessary to support that this descriptor is transferable enough to “broaden” the applicability of the d-band theory.

- We really appreciate your comments. In Mo₂C material, we find that the optimized adsorption free energy and fast HER kinetics can be achieved by manipulating the bond strength between the H 1s orbital and Mo d_z^2 orbital from the viewpoint of traditional orbital overlap theory. And we have calculated the H adsorption free energy (H_{abs}) of Mo atoms on the surface of fcc/hcp GB, and studied its relationship with the Mo d_z^2 orbital energy level (Fig. 6e). Where the numerical fitting over the calculated H_{abs} and the Mo d_z^2 orbital energy level follows the similar function as the eq of $E \propto \frac{f(S, E_{d_z^2})}{E_s - E_{d_z^2}}$ for the bond strength, that is, the adsorption energy varies with the Mo d_z^2 orbital energy level. As a result, the intrinsic HER activity exhibits a strong dependence on the Mo d_z^2 orbital energy level, which rationalizes the role of d_z^2 orbital energy level as one of the most important activity descriptors for HER.

As for the applicability to other materials, we are not sure it can work or not. But that does not mean this work is not valuable. Actually, the applicability of the descriptor can be continuously verified during the researcher's future scientific exploration. Moreover, there are many types of reported descriptors, and not all of them are applicable to all kinds of material, and the applicability of these proposed models are tested by various research groups from the world for decades from the initial inventor. Thus, we can't do that in this simple article, we just give a new ideal and example to stimulate researchers to explore it.

9. Please provide the atomic coordinates of the calculations (CONTCARS), so interested readers can reproduce the work.

➤ We thank the reviewer for the suggestion. We have included the CONTCARs in the supporting information (2).

REVIEWER COMMENTS

Reviewer #1 (Remarks to the Author):

The authors answered all questions, but this manuscript still doesn't fit the criteria of Nature Communications.

Some minor questions:

In line 177 fig.1b, one hydrogen is bonding to two carbons in the bottom left structure model. This is a wrong structure model. Hydrogen cannot do this.

In line 188, from SI fig.4, we know the concentration of nitrogen and oxygen is higher. The authors ignored the effect of nitrogen and oxygen on HER performance.

Reviewer #4 (Remarks to the Author):

In the current version, the authors answered all my questions and addressed the main problems related to the methodology description, making the paper more reproducible. As previously indicated, the research topic is highly relevant, and the work contains a substantial amount of data and nice results that can be interesting for other researchers in the field. The authors improved the methodology and the description of the results and I am inclined to recommend it for publication. I have only some suggestions that could improve the manuscript:

1. While the authors answered about the construction of computational models for the fcc/hcp, fcc/fcc, and hcp/hcp Mo₂C Gbs, a more detailed discussion about the construction of each model, even if included only in the SI, would make the paper more reproducible.

2) The Supplementary Figs 53 and 54 could also indicate some quantitative parameter for the reader, such as the distance of the optimal sites to each GB, to strengthen the argument about the GB importance for the activity.

3) Including a comment about the reasons that make the dilute adsorption model + CHE enough for the description of the experiments would improve the authors arguments.

4) I understand that the authors cannot test the descriptor for other materials in a single article. Indeed, if the descriptor is not applicable to other classes of materials, this would not mean that the work is not valuable. However, I also see that the authors' claim: "As for the applicability to other materials, we are not sure it can work or not." is not described in the paper. Throughout the paper, we see statements such as:

"As a result, the intrinsic HER activity exhibits a strong dependence on the Mo dz₂ orbital energy level, which rationalizes the role of dz₂ orbital energy level as one of the most important activity descriptors for HER."

"Compared with the state-of-the-art descriptors such as band-energy theory and work function that describe the properties of the whole surface⁵⁹, the dz₂ orbital energy level is more relevant to the active site of the catalysts. Therefore, the descriptor dz₂ orbital energy level can be used as a figure of merit for designing transition metal catalyst with well-defined active sites for electrocatalytic reactions, including HER, oxygen evolution reaction (OER), oxygen reduction reaction (ORR), nitrogen reduction reaction (NRR), and carbon dioxide reduction reaction (CO₂RR), etc."

The data rationalize the dz₂ orbital energy level as an important descriptor for one type of material exploring one type of effect. The paper does not show data supporting the other claims about the descriptor. Thus, it would be better to clearly indicate what is really observed and what the authors suggest that should be tested by further studies.

Response to the reviewers

Reviewer #1 (Remarks to the Author):

The authors answered all questions, but this manuscript still doesn't fit the criteria of Nature Communications.

Some minor questions:

1. In line 177 fig.1b, one hydrogen is bonding to two carbons in the bottom left structure model. This is a wrong structure model. Hydrogen cannot do this.

- We are grateful for the reviewer's professional comments. We are sorry that there are still some questions that we are not explained clear in the previous version. Yes, you are right. One hydrogen bonded to two carbons is a wrong structural model. We apologize for the misleading viewing angle of the structural model we presented in the previous manuscript. We have adjusted the structure model to the appropriate view to present the true coordination information, and the corrected **Fig.1b** has been added in the revised manuscript.

More details are provided in revised manuscript (**Fig. 1b**). (Page 9)

Fig. 1. Formation of H-Mo₂C/NG. a Schematic diagram of the synthesis of Mo₂C/NG and H-Mo₂C/NG. **b** Various reaction states of IS_{NP} and IS_{NS} along the reaction pathway.

Inset shows the corresponding reaction ΔH diagram.

2. In line 188, from SI fig.4, we know the concentration of nitrogen and oxygen is higher. The authors ignored the effect of nitrogen and oxygen on HER performance.

- We thank you for your comments. Yes, as revealed in **Supplementary Fig. 4**, the HAADF-STEM image and corresponding EDS mapping prove that Mo, C, N, and O elements overlap quite well and are homogeneously dispersed in H-Mo₂C/NG with nitrogen and oxygen content of 4.50 wt% and 7.49 wt%, respectively (**Supplementary Table 2**). And we do not deny that nitrogen and oxygen in Mo₂C hybrids can affect HER performance, but the grain boundaries in Mo₂C nanosheets are confirmed to be the dominant factor affecting the HER performance.

Previous reports (1, 2) have revealed that the ΔG_H increases close to zero with increasing nitrogen content, which reflects the weakening of Mo–H bonding. However, with excessive N-doping, a positive ΔG_H (0.165 eV) can be observed, indicating that the surface bond H strength becomes too weak to capture H. And the doped nitrogen atom can't modify the HER activity while the adsorption site is far from nitrogen atom, suggesting the number of the active sites remained nearly unchanged. These theoretical results indicate that the enhancement of HER activity with low nitrogen content is not significant. In this work, the atomic percentage of Mo³⁺ (Mo–N) in H-Mo₂C/NG with rich grain boundary is 10.01%, which is slightly higher than that of Mo₂C/NG (8.83%) (**Fig. 4b**); the total nitrogen content of H-Mo₂C/NG (3.90 at%) is lower than that of Mo₂C/NG (4.49 at%) (**Supplementary Table 3**). However, compared with Mo₂C/NG, H-Mo₂C/NG with lower nitrogen content affords sharply enhanced HER activities in both acidic and alkaline electrolytes (**Fig. 5**). **Therefore, the effect of nitrogen in Mo₂C hybrids on HER performance can be ignored.**

Additionally, as shown in **Fig. 4b**, Mo 3d spectra of Mo₂C hybrids are deconvoluted into four typical oxidation states (Mo²⁺, Mo³⁺, Mo⁴⁺, Mo⁶⁺) according to the probable existence of molybdenum species, in which the Mo²⁺ and Mo³⁺ can be ascribed to molybdenum carbides and nitrides, respectively, and the remaining Mo⁴⁺ and Mo⁶⁺ can be attributed to MoO₂ and MoO₃. MoO₂ and MoO₃ are unavoidable for Mo₂C-based materials due to their gradual oxidation at the surface upon exposure to air (3, 4). Previous reports show that surface MoO₃ in Mo₂C is in situ reduced to MoO₂ species during HER (5). As indicated by DFT calculations, the in situ reduced surface with terminal Mo=O moieties has free energy of hydrogen adsorption (ΔG_H) values of –0.236, –0.238, –0.260, and –0.266 eV on Mo sites closer to thermodynamic neutrality, which are better than those of bare Mo₂C (101) (–0.329 eV), MoO₂ (1.131 eV), Mo–O–Mo moieties (–0.421, –0.457, –0.542, and –0.395 eV), O=Mo=O, O=Mo–O–Mo, and Mo–O–Mo–O–Mo moieties (most of ΔG_H values for the models introducing two oxygen atoms are quite negative or positive). Although ΔG_H values for reduced Mo₂C surfaces with terminal Mo=O moieties are closer to thermodynamic neutrality among all possible Mo–O models, these values are not comparable to those at grain boundaries (such as –0.005, –0.010, –0.013, –0.044, 0.035, and 0.038 eV) for our H-Mo₂C/NG

system (Fig. 6e and Supplementary Figs. 53, 54). Meanwhile, experiments show that MoO₂ and MoO₃ are not efficient catalysts for HER (6, 7). In detail, MoO₂ and MoO₃ require extremely large overpotentials of about 430 and 780 mV at pH = 0, and 440 and 660 mV at pH = 14 to reach a current density of 10 mA cm⁻², respectively. **Therefore, the effect of oxygen in Mo₂C hybrids on the HER performance can be ignored, and the dense grain boundaries in Mo₂C nanosheets are confirmed to be the dominant factor affecting the HER performance.**

We are sorry that there are some questions in the previous version. These questions have been corrected in our revised version, and the replies to your comments are given above. Again, we truly appreciate the reviewer's careful review and kind guidance on this work, which is vital for us to improve quality of this manuscript. We wish the revision fulfills your requirements to be published in *Nature Communications*.

References

1. Huang, Y. et al. Fine tuning electronic structure of catalysts through atomic engineering for enhanced hydrogen evolution. *Adv. Energy Mater.* **8**, 1800789 (2018).
2. Wang, Z. et al. Theoretical calculation guided electrocatalysts design: Nitrogen saturated porous Mo₂C nanostructures for hydrogen production. *Appl. Catal. B* **257**, 117891 (2019).
3. Wan, C., Regmi Y. N. & Leonard B. M. Multiple phases of molybdenum carbide as electrocatalysts for the hydrogen evolution reaction. *Angew. Chem. Int. Ed.* **53**, 6407–6410 (2014).
4. Ma, R. et al. Ultrafine molybdenum carbide nanoparticles composited with carbon as a highly active hydrogen-evolution electrocatalyst. *Angew. Chem. Int. Ed.* **54**, 14723–14727 (2015).
5. He, L. et al. Molybdenum carbide-oxide heterostructures: In situ surface reconfiguration toward efficient electrocatalytic hydrogen evolution. *Angew. Chem. Int. Ed.* **59**, 3544–3548 (2020).
6. Li, J.-S. et al. Coupled molybdenum carbide and reduced graphene oxide electrocatalysts for efficient hydrogen evolution. *Nat. Commun.* **7**, 11204 (2016).
7. Vrubel, H. & Hu, X. Molybdenum boride and carbide catalyze hydrogen evolution in both acidic and basic solutions. *Angew. Chem. Int. Ed.* **51**, 12703–12706 (2012).

Reviewer #4 (Remarks to the Author):

In the current version, the authors answered all my questions and addressed the main problems related to the methodology description, making the paper more reproducible. As previously indicated, the research topic is highly relevant, and the work contains a substantial amount of data and nice results that can be interesting for other researchers in the field. The authors improved the methodology and the description of the results and I am inclined to recommend it for publication. I have only some suggestions that

could improve the manuscript:

- We thank the reviewer for this very positive review. The main points have been largely corrected in our revised version, and all changes are highlighted in blue throughout the revision. We wish the revision fulfills your requirements to be published in *Nature Communications*.

1. While the authors answered about the construction of computational models for the fcc/hcp, fcc/fcc, and hcp/hcp Mo₂C GBs, a more detailed discussion about the construction of each model, even if included only in the SI, would make the paper more reproducible.

- We really appreciate your suggestions for improving the manuscript. We have added the discussion in the revised supporting information: “According to the HAADF-STEM images of Mo₂C NSs, the computational models of Mo₂C with fcc/hcp, fcc/fcc, and hcp/hcp GBs were constructed, respectively. Specifically, **Fig. 3e** shows the superstructure of fcc/hcp heterophase GB in Mo₂C nanosheets. Along the fcc/hcp heterophase edges, both hcp phase with “ABAB” stacking and fcc phase with “ABCABC” stacking are observed along the close-packed [110]_f/[112̄ 0]_h direction, which clearly reveals that the heterophase Mo₂C GB is generated through the chemical connection of fcc(111) and hcp(002) slabs. By analogy, we used the software of VESTA to successfully construct fcc(111)/fcc(010) (**Supplementary Fig. 25**), hcp(002)/hcp(110), hcp(10 1̄)/hcp(110), hcp(010)/hcp(001), and hcp(010)/hcp(101) models (**Fig. 3a, c** and **Supplementary Fig. 22**) to simulate the fcc/hcp, fcc/fcc, and hcp/hcp Mo₂C GBs in actual material. The structural models obtained by subsequent optimization screening were shown in **Supplementary Figs. 53, 54.**”
(Page 69-70)

2. The Supplementary Figs 53 and 54 could also indicate some quantitative parameter for the reader, such as the distance of the optimal sites to each GB, to strengthen the argument about the GB importance for the activity.

- We really appreciate your suggestions for improving the manuscript. We have added the distance of the optimal site to each GB in **Supplementary Figs. 53, 54**. More details are provided in revised supporting information (**Supplementary Figs. 53, 54**). (Page 57-58)

Supplementary Fig. 53. Atomic structure of fcc/hcp GB. Mo atoms with optimized H_{ads} at fcc/hcp GB are highlighted by yellow circles. The cyan and dark gray balls represent Mo and C atoms, respectively. The vicinity of the fcc/hcp GB exhibits optimal H adsorption free energies (such as -0.005 , -0.010 , -0.013 , and 0.058 eV), which are superior to those of sites far from the GBs (such as -0.324 , -0.358 , -0.428 , and -0.517 eV) (Fig. 6e).

Supplementary Fig. 54. Atomic structure of fcc/fcc and hcp/hcp GBs. **a** fcc/fcc and **b-e** hcp/hcp GBs. **f** H_{ads} diagram. The dashed line is the H_{ads} of Pt (111) (-0.121 eV). The data points in the shaded area in **f** are generated using the adsorption sites indicated by the yellow circles. The cyan and dark gray balls represent Mo and C atoms, respectively.

3. Including a comment about the reasons that make the dilute adsorption model + CHE enough for the description of the experiments would improve the authors arguments.

➤ We appreciate your suggestions for improving the quality of this article. We have added the comment in the revised supporting information: “In 1958, Parson first

pointed out that a maximum exchange current density would be attained when the hydrogen adsorption free energy was close to thermoneutral (~ 0 eV)³⁷. Norskov et al. plotted hydrogen adsorption free energy from DFT calculations versus experimental HER exchange current density³⁸, and observed a volcano-shaped variation with the peak position close to platinum—note that platinum was the state-of-the-art HER catalyst with an almost zero overpotential and small Tafel slope. This volcano plot was further consummated in other studies in the literatures^{39, 40}. These reports suggest that hydrogen adsorption free energy can be exploited as an effective descriptor in the design, engineering and optimization of HER catalysts. In this work, the H is adsorbed on top of the Mo atoms, which is the same as that on the surface of Pt. Therefore, the computational hydrogen electrode (CHE) method was used to calculate the hydrogen adsorption free energy of surface Mo atoms in Mo₂C. In addition, dilution models consistent with experiment were employed at very small applied potential.” (Page 68-69)

4. I understand that the authors cannot test the descriptor for other materials in a single article. Indeed, if the descriptor is not applicable to other classes of materials, this would not mean that the work is not valuable. However, I also see that the authors’ claim: “As for the applicability to other materials, we are not sure it can work or not.” is not described in the paper. Throughout the paper, we see statements such as:

“As a result, the intrinsic HER activity exhibits a strong dependence on the Mo d_z^2 orbital energy level, which rationalizes the role of d_z^2 orbital energy level as one of the most important activity descriptors for HER.”

“Compared with the state-of-the-art descriptors such as band-energy theory and work function that describe the properties of the whole surface⁵⁹, the d_z^2 orbital energy level is more relevant to the active site of the catalysts. Therefore, the descriptor d_z^2 orbital energy level can be used as a figure of merit for designing transition metal catalyst with well-defined active sites for electrocatalytic reactions, including HER, oxygen evolution reaction (OER), oxygen reduction reaction (ORR), nitrogen reduction reaction (NRR), and carbon dioxide reduction reaction (CO₂RR), etc.”

The data rationalize the d_z^2 orbital energy level as an important descriptor for one type of material exploring one type of effect. The paper does not show data supporting the other claims about the descriptor. Thus, it would be better to clearly indicate what is really observed and what the authors suggest that should be tested by further studies.

➤ We appreciate your comments and suggests. We are very sorry for some inaccurate statements in the previous version, such as: “As a result, the intrinsic HER activity exhibits a strong dependence on the Mo d_z^2 orbital energy level, which rationalizes the role of d_z^2 orbital energy level as one of the most important activity descriptors for HER.” and “Compared with the state-of-the-art descriptors such as band-energy theory and work function that describe the properties of the whole surface⁵⁹, the d_z^2 orbital energy level is more relevant to the active site of the catalysts. Therefore, the descriptor d_z^2 orbital energy level can be used as a figure of merit for designing transition metal catalyst with well-defined active sites for electrocatalytic reactions,

including HER, oxygen evolution reaction (OER), oxygen reduction reaction (ORR), nitrogen reduction reaction (NRR), and carbon dioxide reduction reaction (CO₂RR), etc.”

In our case, the numerical fitting over the calculated H_{ads} and the Mo d_z^2 orbital energy level follows the similar function as the of $E \propto \frac{f(S, E_{d_z^2})}{E_s - E_{d_z^2}}$ for the bond strength. That is, the intrinsic HER activity exhibits a strong dependence on the Mo d_z^2 orbital energy level. We have corrected the inaccurate statements in revised manuscript: “As a result, the intrinsic HER activity exhibits a strong dependence on the Mo d_z^2 orbital energy level, which rationalizes the role of d_z^2 orbital energy level as one of the most important activity descriptors for HER in Mo₂C systems. Compared with the state-of-the-art descriptors such as band-energy theory and work function that describe the properties of the whole surface⁵⁹, the d_z^2 orbital energy level is more relevant to the active site of the catalysts. Therefore, the descriptor d_z^2 orbital energy level could be used as a figure of merit for designing transition metal catalyst with well-defined active sites for electrocatalytic HER reaction.” (Page 32)